# ONE FOR ALL: ZERO-SHOT CROSS-HARDWARE PERFORMANCE MODELING WITH LLMS FOR TENSOR PROGRAM TUNING

## ABSTRACT

Tensor program tuning is critical for inference acceleration of deep neural networks (DNNs), especially Large Language Models (LLMs). Yet its effectiveness hinges on cost models for accurate performance estimation. Existing cost models rely on manually designed hardware-specific features and extensive profiling data. Thus they suffer from high development costs, poor efficiency, and limited generalization, and become to a significant bottleneck in the face of rapidly evolving models and hardware. In this paper, we propose LLMTuner, a novel framework enabling LLMs to analyze tensor program execution behaviors and accurately estimate tensor program performance across diverse hardware. LLMTuner introduces a coarse-to-fine process: a lightweight LLM-based classifier first filters out suboptimal programs, then a finetuned LLM infers multi-dimensional execution behavior scores to predict latency across different hardware. Experiments demonstrate that LLMTuner significantly improves estimation accuracy by up to 64.8%, compared with general-purpose LLMs and other cost models on benchmark datasets across 6 CPU and 5 GPU platforms. It can even accurately estimate performance on unseen hardware, achieving 49.2% accuracy improvement over other cost models. For practical DNN and LLM tuning tasks, compared with other cost models, LLMTuner could discover superior program performance ($1.47\times$) with up to $3.27\times$ tuning efficiency. Moreover, LLMTuner with finetuned lightweight LLMs reduces the estimation time by over $30\times$ compared to DeepSeek R1.

## 1 INTRODUCTION

In recent years, the rapid advancement of deep neural networks (DNNs), particularly large language models (LLMs), has created substantial demand for optimizing inference latency on diverse hardware (Maslej et al., 2025; Salaria et al., 2025). To reduce model inference latency, tensor program tuning serves as a key technique in deep learning compilers for discovering efficient implementations of models across different platforms (Chen et al., 2018b). As a core component of tensor program tuning, the cost model utilizes trained neural networks to predict program latency and select high-performance candidates for subsequent tuning. This eliminates the need for exhaustive hardware measurements, as shown in Figure 1 (a). The cost model guides tuning directions such as loop unrolling and vectorization, which are crucial for both tuning efficiency and final performance.

The traditional approach (Marculescu et al., 2018; Baghdadi et al., 2021; Zhai et al., 2023; Qiao et al., 2025) to build cost models involves a hand-crafted set of target hardware-specific program features (e.g., the number of operations and loop lengths). These features are then used to train models on performance data collected either online or offline. As shown in Figure 1(b), this paradigm not only requires substantial expert effort and long training times (ranging from hours to days) but also yields models that generalize poorly to new hardware (accuracy dropping from 89.2% to 62.7%). The high cost, low efficiency, and poor generalization remain major bottlenecks for applying tensor program tuning to rapidly evolving DNN models and hardware platforms.

The fundamental limitation of the aforementioned paradigm is that handcrafted static program features cannot generalize across hardware, since their performance implications vary from one platform to another. In contrast, modeling the execution behavior of tensor programs on hardware (e.g.,

Figure 1: (a) Cost models play a central role in tensor program tuning. (b) Traditional cost models are trained separately for each hardware platform and thus lack generalization. (c)Vanilla LLMs cannot understand and analyze code performance. (d) Our LLMTuner captures the unified execution behavior of programs across diverse hardware, enabling effective cross-platform generalization.

compute unit utilization and memory access patterns) captures performance-relevant characteristics that transfer across devices. Motivated by this observation, we propose a new paradigm that builds generalizable cost models by analyzing unified execution behaviors rather than hardware-specific static features. Unlike traditional machine learning models, LLMs possess powerful capabilities in high-level program understanding, reasoning, and in-context learning, holding the potential to realize this novel paradigm. However, general-purpose LLMs currently lack the ability to understand and analyze code performance in relation to hardware-specific characteristics, as shown in Figure 1(c). Therefore, the key technical challenge addressed in this paper is how to enable LLMs to automatically analyze unified execution behaviors of tensor programs for cross-hardware performance modeling.

In this paper, we propose LLMTuner, a novel two-stage coarse-to-fine framework enabling LLMs to analyze tensor program execution behaviors and accurately estimate tensor program performance across diverse hardware, as shown in Figure 1(d). In the first coarse filtering stage, a trained lightweight-LLM-based classifier rapidly filters out clearly suboptimal programs. The second fine selection stage takes the surviving programs along with target hardware factors as input, leveraging a finetuned LLM to infer a set of multi-dimensional quantitative performance scores, which serve as a hardware-agnostic intermediate representation. These scores are then fed into a general regression model to predict the actual latency, guiding the selection of the most promising programs for the next tuning iteration. Such a mechanism ensures cross-hardware performance estimation accuracy while significantly improving the efficiency of large-scale tensor program tuning.

Overall, the main contributions of this paper are as follows:

- We propose a cross-hardware cost modeling paradigm for tensor program tuning, leveraging LLMs to analyze unified program execution behaviors and overcome the limitations of traditional cost models.
- We design a coarse-to-fine framework and fine-tune LLMs to enable accurate and efficient tensor program performance estimation on diverse hardware.
- Experiments demonstrate that LLMTuner significantly improves estimation accuracy by up to 64.8%, compared with general-purpose LLMs and other cost models on benchmark datasets across 6 CPU and 5 GPU platforms. It can even accurately estimate performance on unseen hardware, achieving 49.2% accuracy improvement over other cost models. For practical DNN and LLM tuning tasks, compared with other cost models, LLMTuner could exploit superior program performance ($1.47\times$) with up to $3.27\times$ tuning efficiency. Moreover, LLMTuner with finetuned lightweight LLMs reduces the estimation time by over $30\times$ compared to DeepSeek R1.

## 2 RELATED WORKS

**Cost Model in Deep Learning Compilers.** Search-based deep learning compilers rely on cost models to guide program optimization. As shown in Table 1, early systems combined hand-crafted

features with classical machine learning models such as XGBoost, which limited their generalization ability and required retraining on each new hardware platform. Subsequent works introduced neural predictors (e.g., MLPs and LSTMs) or leveraged transformer-based encoders to better capture structured program information. More recent efforts employed multi-task learning, pruning strategies, and other techniques to improve efficiency on new platforms, but still rely on hand-crafted features and need to be finetuned for new platform adaptation. In contrast, our approach removes the dependence on hand-crafted features. By directly modeling the unified execution behavior by LLM reasoning, our method achieves efficient optimization and robust cross-platform generalization without the need for retraining or adaptation.

Table 1: Comparison of representative cost models in search-based deep learning compilers.

| Related Works | Feature Extraction | Model | Multi-Platform Support | New-Platform Adaptation |
|---|---|---|---|---|
| AutoTVM (Chen et al., 2018b) | Hand-crafted | XGBoost | ✗ | ✗ |
| Ansor (Zheng et al., 2020a) | Hand-crafted | XGBoost | ✗ | ✗ |
| MetaSchedule (Shao et al., 2022) | Hand-crafted | XGBoost | ✗ | ✗ |
| Heron (Bi et al., 2023) | Hand-crafted | XGBoost | ✗ | ✗ |
| TenSetMLP (Zheng et al., 2020b) | Hand-crafted | MLP | ✗ | ✗ |
| TIRAMISU (Baghdadi et al., 2021) | Hand-crafted | LSTM | ✗ | ✗ |
| TLP (Zhai et al., 2023) | Language Tokens | Transformer | ✗ | ✗ |
| MTL-TLP (Zhai et al., 2023) | Language Tokens | Transformer | ✓ 5 CPUs + 2 GPUs | ✗ Need Finetune |
| Moses (Zhao et al., 2023) | Hand-crafted | MLP | ✓ 3 GPUs | ✗ Need Finetune |
| Pruner (Qiao et al., 2025) | Hand-crafted | Transformer | ✗ | ✗ |
| MoA-Pruner (Qiao et al., 2025) | Hand-crafted | Transformer | ✓ 5 GPUs | ✗ Need Finetune |
| **Ours (LLMTuner)** | Fine-tuned LLM Reasoning | Lightweight LLM | ✓ 6 CPUs + 5 GPUs | ✓ |

**LLMs for Program Understanding.** Recent work has demonstrated that LLMs can capture high-level program semantics for performance-related tasks. For example, LLMPerf (Nguyen-Nhat et al., 2024) estimates OpenCL kernel runtimes from static code, another study (Bolet et al., 2025) classifies GPU kernels as compute- or memory-bound from source code, and HPC-Coder (Nichols et al., 2024) compares the relative performance of C++ program pairs. While these efforts highlight the ability of LLMs to understand program behavior, they remain limited to single-hardware tasks and cannot combine program and hardware characteristics to analyze performance. In contrast, our work targets tensor program tuning, a more challenging and lower-level task. And we enable LLMs to perform structured reasoning for generalizable and hardware-aware performance analysis.

## 3 PRELIMINARY

In this section, we provide a detailed introduction to the key components involved in tensor program tuning. As illustrated in Figure 1 (a), deploying high-level DNN models (e.g., PyTorch) onto a target hardware platform $h$ requires translating abstract computational graphs $\mathcal{G}$ into low-level *tensor programs* that are specialized for the hardware's execution characteristics. A tensor program $p \in \mathcal{P}_g$ is a loop-level representation of a compute-intensive subgraph $g \in \mathcal{G}$ (e.g., Convolution-batchnorm-relu subnetwork in DNN). The objective of tensor program tuning is to find an optimized implementation $p^* = \arg\min_{p \in \mathcal{P}_g} T(p, h)$, where $T(p, h)$ denotes the execution time (measured or predicted) of program $p$ on hardware $h$, and $\mathcal{P}_g$ is the search space of candidate tensor programs for $g$.

To explore $\mathcal{P}_g$, a *program generator* $\mathcal{G}_p$ is employed to schedule various code transformations (e.g., loop tiling, vectorization, memory reuse). This process defines a large combinatorial search space:

$$\mathcal{P}_g = \{ p \mid p = \mathcal{G}_p(g, \theta), \ \theta \in \Theta \}, \tag{1}$$

where $\Theta$ is the parameter space of scheduling decisions. Search algorithms such as Monte Carlo Tree Search (MCTS) (Chaslot, 2010) or Genetic Algorithms (GA) (Kramer, 2017) iteratively propose new candidates $p \in \mathcal{P}_g$ by refining $\theta$ based on previous evaluations.

Since evaluating all candidates on hardware is prohibitively expensive, a *cost model* $\hat{T}(p, h)$ is used to predict the latency of each candidate without actual execution. The role of $\hat{T}$ is twofold: it eliminates unpromising candidates with large predicted latency and provides performance feedback to the generator, guiding the search toward promising regions of $\mathcal{P}_g$. By iteratively alternating between candidate generation and cost-model-based prediction, the system converges to high-performance

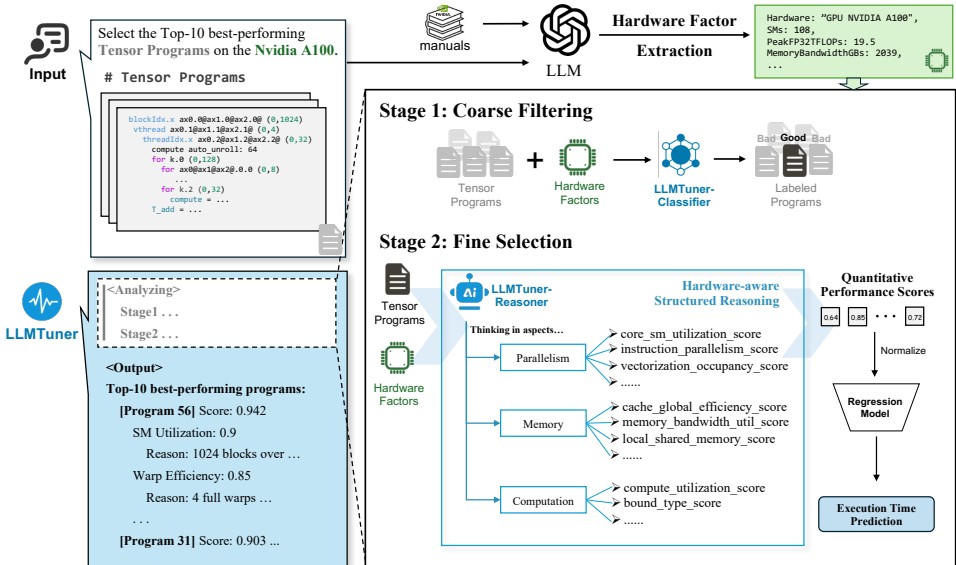

Figure 2: Overview of the LLMTuner pipeline. LLMTuner first extracts key hardware factors, then uses LLMTuner-Classifier for coarse candidate filtering, and finally applies LLMTuner-Reasoner for performance analysis and execution time prediction.

implementations $p^*$ for each $g \in \mathcal{G}$ and the target hardware $h$. The overall algorithm is described in Algorithm 1.

# 4 LLMTUNER

In this section, we introduce the proposed **LLMTuner** framework, which efficiently selects high-performance tensor programs through a two-stage *coarse-to-fine* process. As illustrated in Figure 2, LLMTuner takes a batch of tensor programs and the target hardware name as input. It first performs automatic hardware factors extraction, then the **LLMTuner-Classifier** uses the tensor programs and hardware factors to conduct coarse filtering, followed by the fine selection by **LLMTuner-Reasoner**. The final output is a set of top-performing candidate programs.

## 4.1 AUTOMATED HARDWARE FACTOR EXTRACTION

For each target hardware, our framework uses an LLM to automatically extract key specifications from official manuals. Given a hardware name (e.g., "NVIDIA A100" or "Intel Xeon Platinum 8272CL"), the LLM retrieves a predefined set of attributes, such as the number of SMs, warp size, and peak FLOPs for GPUs, or core/thread counts, SIMD/AVX width, and memory bandwidth for CPUs. This automated approach enables efficient, structured, and scalable collection of hardware factors without manual effort.

## 4.2 COARSE FILTERING VIA LLMTUNER-CLASSIFIER

To efficiently identify high-performing tensor programs, we adopt a two-stage design. As shown in Stage 1 of Figure 2, the coarse filtering rapidly eliminates candidates that are unlikely to achieve high performance, thereby reducing the computational cost of subsequent fine-grained selection. This stage is realized using the *LLMTuner-Classifier*. The LLMTuner-Classifier employs a lightweight LLM augmented with a single-layer classification head applied to the last hidden state. The input to the model combines the program code, target hardware factors, and a classification instruction, which is tokenized and fed into the LLM. Since LLMs support long-context inputs, the LLMTuner-Classifier can efficiently scale to evaluating hundreds of candidates. The classifier is trained on a large-scale tensor program performance dataset TenSet Zheng et al. (2021), where candidates are labeled according to their measured latency (training details are provided in Section 4.4).

### 4.3 FINE SELECTION VIA LLMTUNER-REASONER

The fine selection stage aims to provide accurate and interpretable performance estimation for the subset of candidates that pass the initial coarse filtering. We fine-tune LLMs to perform hardware-aware structured reasoning, capturing the program–hardware execution behaviors. These behaviors are encoded as multi-dimensional performance scores, which serve as a cross-platform intermediate representation. The scores are then fed into a lightweight regression model to predict actual execution time.

#### 4.3.1 HARDWARE-AWARE STRUCTURED REASONING

As shown in Stage 2 of Figure 2, Hardware-aware Structured Reasoning refers to a principled approach that integrates explicit hardware knowledge into the structured evaluation of program candidates.

The "Hardware-aware" implies that reasoning is conditioned on hardware-specific factors (e.g., SM count, peak FLOPs, memory bandwidth of the Nvidia A100 GPU). These factors provide explicit, human-interpretable context for downstream analysis.

The "Structured reasoning" is implemented by decomposing the reasoning process into three core dimensions: **Parallelism**, **Memory**, and **Computation**. Each dimension is represented by a set of quantitative metrics that capture the execution behavior of tensor programs on the target hardware.

**Parallelism** metrics assess how effectively the program utilizes the available computational resources. For CPUs, this includes evaluation of multi-core and multi-thread distribution, instruction-level parallelism (ILP), SIMD vectorization efficiency, load balancing, and synchronization overheads. In the context of GPUs, parallelism metrics capture streaming multiprocessor (SM) coverage, warp divergence, occupancy rates, block and warp distribution balance, and the cost of thread-level synchronization.

**Memory** metrics capture efficiency in accessing and utilizing hierarchical memory systems. These measurements include cache hit rates, memory bandwidth utilization, shared and local memory usage, register pressure, bank conflicts, and the exploitation of temporal and spatial locality. The metric suite reflects both CPU-centric issues, such as NUMA locality and cache line utilization, and GPU-specific phenomena, including global memory coalescing and shared memory conflicts.

**Computation** metrics evaluate the extent to which arithmetic resources are effectively utilized, as well as the balance of instruction types across the program. Key indicators include the utilization of compute units—such as ALU, FPU, or specialized Tensor cores—the impact of loop unrolling on ILP or warp utilization, differentiation between memory-bound and compute-bound execution phases, and the balance of instruction mixes.

To enable LLMs to automatically analyze the execution behaviors described above, we collect a large set of high-quality prompt–response pairs, each annotated with detailed reasoning traces and multi-dimensional performance scores. The LLMTuner-Reasoner is then fine-tuned on this dataset (see Section 4.4 for training details).

#### 4.3.2 MAPPING REASONING FEATURES TO EXECUTION TIME

The LLMTuner-Reasoner produces a set of multi-dimensional alignment scores for each candidate, directly reflecting the efficiency of its execution on the target hardware. These scores are normalized and fed into a regression model, which is trained to predict actual execution time. The regression model learns to map semantic assessments to quantitative latency estimates. By modeling the execution behavior of programs on target hardware, rather than merely learning a direct mapping from program to execution time as in prior work, this approach facilitates robust generalization to previously unseen devices.

### 4.4 DATASET PREPARATION AND TRAINING DETAILS

In this section, we introduce the dataset preparation and training procedures for each part of LLM-Tuner. Due to page limit, more details are provided in the Appendix C.

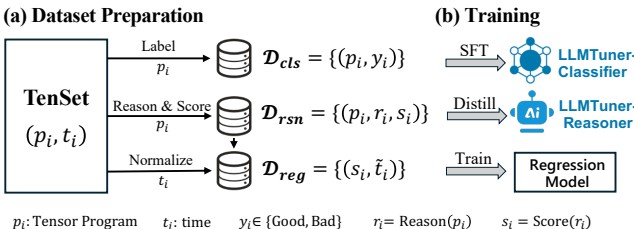

Figure 3: Illustration of Data Preparation and Training.

**Dataset Preparation.** As shown in Figure 3, from the raw tensor program set $\mathcal{P} = \{(p_i, t_i)\}_{i=1}^{N}$ of TenSet, where $p_i$ denotes a tensor program and $t_i$ its execution time, we construct three datasets to train different components of LLMTuner:

$$\mathcal{D}_{\text{cls}} = \{(p_i, y_i) \mid y_i \in \{\text{Good, Bad}\}\} \tag{2}$$

$$\mathcal{D}_{\text{rsn}} = \{(p_i, r_i, s_i) \mid r_i = \text{Reason}(p_i, t_i),\ s_i = \text{Score}(r_i)\} \tag{3}$$

$$\mathcal{D}_{\text{reg}} = \{(s_i, \tilde{t}_i) \mid s_i = \text{Score}(p_i),\ \tilde{t}_i = \text{Normalize}(t_i)\} \tag{4}$$

Specifically, $\mathcal{D}_{\text{cls}}$ is constructed for SFT of the LLMTuner-Classifier. For each subgraph in TenSet, the top $k$ candidate programs with the lowest latency are labeled as "good", while the remaining are labeled as "bad". $\mathcal{D}_{\text{rsn}}$ is curated for distillation training of the LLMTuner-Reasoner. It contains approximately 100K high-quality prompt-response pairs, each annotated with detailed reasoning traces and program–hardware alignment scores generated by DeepSeekR1-671B. During the generation, real program execution time is provided as a hint to guide the scoring and reasoning process. $\mathcal{D}_{\text{reg}}$ is used to train the regression model, consisting of pairs of alignment scores and corresponding ground-truth latency. All latency values are normalized to $[0, 1]$.

**Training Details.** We adopt a lightweight LLM as the base model for the LLMTuner-Classifier, extending it with a single-layer classification head for binary prediction. To address class imbalance, we downsample the majority class and train with class-weighted MSE loss. For the LLMTuner-Reasoner, we finetune a pre-trained LLM with high-quality prompt-response pairs from $\mathcal{D}_{\text{rsn}}$ using LoRA Hu et al. (2022). Finally, the multi-layer MLP regression model is optimized using a combination of ranking loss and MSE loss.

## 5 Experiments

To verify the effectiveness of LLMTuner, we conduct experiments on tensor program tuning tasks of various hardware and DNNs, comparing its performance against previous cost models. We further evaluate the performance estimation accuracy on widely used benchmarks across a variety of both seen and unseen hardware platforms compared with various general-purpose LLMs and cost models.

### 5.1 Experiment Setup

**Datasets.** TenSet (Zheng et al., 2021) is a widely used benchmark for cost model learning, composed of tasks from 120 neural networks(e.g., ResNet, Bert, DenseNet) across 6 diverse hardware platforms: Intel Platinum 8272, Intel E5 2673, AMD EPYC 7452, ARM Graviton2, NVIDIA K80, and NVIDIA T4. Each task includes up to 4,000 tensor programs with measured latency. Following the standard split, subgraphs from five networks (e.g., ResNet-50, BERT-base) serve as the test set, while the rest are used for training.

To further evaluate cross-hardware generalization, we curated a large-scale dataset of program–latency pairs collected from five diverse modern hardware platforms, including Intel Platinum 8575C, ARM Cortex-A76, NVIDIA A100, NVIDIA H100, and NVIDIA RTX 4070, covering both old tasks and new tasks such as ViT and GPT-2. Details are provided in Appendix B.

**Implementation Details.** For LLM-Classifier training, we adopt Qwen2.5-0.5B-Instruct as the base model. For LLM-Reasoner training, we fine-tune Qwen2.5-7B-Instruct using LoRA with the reasoning data from DeepSeek-R1. The regression model is implemented as a 3-layer MLP with hidden dimensions of 512, 256, and 128. All training was conducted on NVIDIA A100 GPUs.

**Evaluation Metrics.** We adopt the Top-1 and Top-5 accuracy score (following TenSet), which compare how close the best latency among the top-$k$ candidates selected by the cost model is to the overall best latency across all candidates. (Details definition see Eq. 5 in Appendix.)

**Baselines.** (1) State-of-the-art LLMs: Gemini (Gemini et al., 2023), GPT (Hurst et al., 2024), Claude (Anthropic, 2025), Qwen (Qwen, 2024) and DeepSeek (Guo et al., 2025). (2) Traditional Cost Models: Ansor (Zheng et al., 2020a), TenSetMLP (Zheng et al., 2021), TLP (Zhai et al., 2023), and Pruner (Qiao et al., 2025).

## 5.2 APPLYING LLMTUNER TO ONLINE TENSOR PROGRAM TUNING

To evaluate the effectiveness of LLMTuner on tensor program tuning, we integrate LLMTuner into the auto-tuning framework TVM (Chen et al., 2018a) and perform extensive experiments on unseen CPU and GPU in training dataset. We evaluate tuning on four models, including DeepSeek-V3, Qwen2-7B, ResNet50, and BERT-base, using a batch size of 1 and a sequence length of 128 (or image size=224). The DeepSeek-V3 is a simplified version that preserves the core Transformer+MoE architecture while replacing unsupported operations to ensure compatibility with TVM Relay.

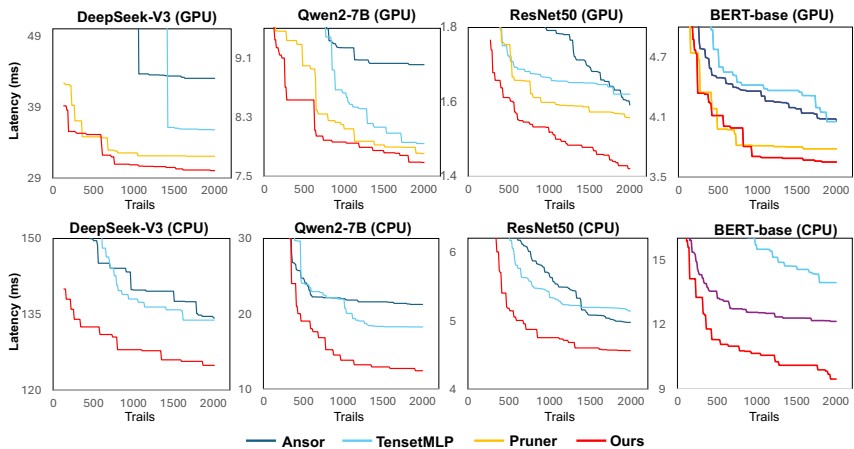

Figure 4: Tuning curves of different methods on NVIDIA A100 GPU and Intel Platinum 8575 CPU.

**LLMTuner can discover superior program performance with fewer tuning steps across diverse tasks on unseen CPU and GPU platforms consistently.** As illustrated in Figure 4, on both NVIDIA A100 GPU and Intel Platinum 8575 CPU platforms, LLMTuner attains superior tuning results with fewer trials compared to established baselines such as Ansor, TenSetMLP, and Pruner. For example, when tuning Qwen2-7B on CPU, it achieves 1.47× final performance compared with TensorMLP with up to 3.27× tuning efficiency. This consistent advantage holds across diverse workloads, which can be attributed to LLMTuner's strong generalization ability. By leveraging large-scale pretraining for unified program execution behavior reasoning, LLMTuner directly adapts to new tuning tasks and hardware, thus efficiently exploring the search space. These results highlight the practical benefits of LLMTuner for more effective tensor program tuning scenarios.

## 5.3 PERFORMANCE ESTIMATION ACCURACY ON TENSET BENCHMARK

To examine the prediction accuracy of LLMTuner, we evaluate it on the TenSet benchmark, comparing it against baselines including state-of-the-art LLMs and traditional cost models. Table 2 reports the overall Top-1 and Top-5 performance.

**LLMTuner enables accurate performance estimation beyond existing LLMs and cost models.** Our results demonstrate that current few-shot LLMs struggle to predict program latency accurately, significantly underperforming both traditional cost models and our proposed approach. For example, on the Intel Platinum 8272 platform, the best LLM baseline (Gemini 2.5 Pro, with 0.287 Top-1 score) lags far behind the TenSetMLP cost model (Top-1: 0.875) and LLMTuner (Top-1: 0.924). Similar trends are observed across other hardware platforms, where even large-scale models like DeepSeek-R1-671B fail to outperform smaller models such as Qwen2.5-7B-Instruct, revealing a

Table 2: Overall Performance Comparison on the TenSet. The best results are highlighted in **bold**, the second-best results are underlined. '-' refers to the failure of Pruner on CPU platforms.

| Model | Platinum 8272 | | E5 2673 | | EPYC 7452 | | Graviton2 | | NVIDIA K80 | | NVIDIA T4 | |
|---|---|---|---|---|---|---|---|---|---|---|---|---|
| | Top1 | Top5 | Top1 | Top5 | Top1 | Top5 | Top1 | Top5 | Top1 | Top5 | Top1 | Top5 |
| **Cost Model** | | | | | | | | | | | | |
| TenSetMLP | 0.875 | 0.953 | 0.833 | 0.898 | 0.851 | 0.918 | 0.780 | 0.905 | 0.908 | 0.963 | 0.876 | 0.953 |
| TLP | 0.919 | 0.971 | 0.894 | 0.963 | 0.906 | 0.949 | **0.821** | **0.923** | 0.906 | 0.974 | 0.885 | 0.925 |
| Pruner | - | - | - | - | - | - | - | - | 0.897 | 0.969 | 0.892 | **0.962** |
| **LLM Few-shot** | | | | | | | | | | | | |
| Gemini 2.5 Flash | 0.113 | 0.251 | 0.105 | 0.246 | 0.101 | 0.239 | 0.099 | 0.232 | 0.187 | 0.358 | 0.214 | 0.372 |
| Gemini 2.5 Pro | 0.287 | 0.323 | 0.191 | 0.402 | 0.183 | 0.395 | 0.179 | 0.388 | 0.346 | 0.453 | 0.380 | 0.470 |
| GPT-4o-mini | 0.091 | 0.215 | 0.085 | 0.297 | 0.081 | 0.291 | 0.079 | 0.284 | 0.134 | 0.313 | 0.160 | 0.320 |
| GPT-4o | 0.189 | 0.327 | 0.176 | 0.314 | 0.171 | 0.308 | 0.168 | 0.301 | 0.271 | 0.451 | 0.310 | 0.470 |
| Claude-3.7-Sonnet | 0.201 | 0.335 | 0.291 | 0.326 | 0.284 | 0.319 | 0.279 | 0.312 | 0.250 | 0.399 | 0.279 | 0.415 |
| DeepSeekV3-671B | 0.082 | 0.186 | 0.075 | 0.179 | 0.072 | 0.173 | 0.070 | 0.168 | 0.122 | 0.267 | 0.134 | 0.296 |
| DeepSeekR1-671B | 0.155 | 0.355 | 0.138 | 0.348 | 0.134 | 0.342 | 0.172 | 0.335 | 0.195 | 0.481 | 0.220 | 0.495 |
| Qwen2.5-7B-Inst | 0.111 | 0.276 | 0.135 | 0.274 | 0.194 | 0.283 | 0.055 | 0.202 | 0.137 | 0.346 | 0.162 | 0.330 |
| ↪ **LLMTuner** | **0.924** | **0.973** | **0.896** | **0.972** | **0.917** | **0.957** | 0.815 | 0.911 | **0.919** | **0.976** | **0.917** | 0.953 |
| | ↑81.3% | ↑69.7% | ↑76.1% | ↑69.8% | ↑72.3% | ↑67.4% | ↑76.0% | ↑70.9% | ↑78.2% | ↑63.0% | ↑75.5% | ↑62.3% |

Table 3: Generalization Performance on the Unseen Hardware. All models are only pretrained on TenSet. The best results are highlighted in **bold**, while the second-best results are underlined. '-' refers to the failure of Pruner on CPU platforms.

| Model | Platinum 8575C | | Cortex-A76 | | NVIDIA A100 | | NVIDIA H100 | | NVIDIA RTX4070 | |
|---|---|---|---|---|---|---|---|---|---|---|
| | Top 1 | Top 5 | Top 1 | Top 5 | Top 1 | Top 5 | Top 1 | Top 5 | Top 1 | Top 5 |
| TenSetMLP | 0.271 | 0.326 | 0.330 | 0.444 | 0.320 | 0.429 | 0.219 | 0.367 | 0.492 | 0.613 |
| TLP | 0.290 | 0.412 | 0.300 | 0.510 | 0.497 | 0.673 | 0.376 | 0.511 | 0.656 | 0.776 |
| Pruner | - | - | - | - | 0.506 | 0.642 | 0.495 | 0.561 | 0.627 | 0.783 |
| DeepSeek R1 671B | 0.118 | 0.278 | 0.102 | 0.319 | 0.141 | 0.388 | 0.201 | 0.341 | 0.212 | 0.452 |
| Qwen2.5-7B-Inst | 0.124 | 0.267 | 0.119 | 0.296 | 0.132 | 0.301 | 0.193 | 0.297 | 0.222 | 0.375 |
| ↪ **LLMTuner** | **0.794** | **0.886** | **0.803** | **0.892** | **0.812** | **0.902** | **0.787** | **0.874** | **0.857** | **0.931** |
| | ↑67.0% | ↑61.9% | ↑68.4% | ↑59.6% | ↑68.0% | ↑60.1% | ↑59.4% | ↑57.7% | ↑63.5% | ↑55.6% |

performance bottleneck in current LLMs. In contrast, LLMTuner achieves state-of-the-art results by enabling LLMs with structured, hardware-aware reasoning. On the NVIDIA Tesla K80, for instance, LLMTuner reaches a Top-1 score of 0.919, surpassing both the gpt-4o (Top-1: 0.271) by 64.8% and the traditional cost model (Top-1: 0.908). These findings highlight the effectiveness of our method and the necessity of training LLMs with performance-aligned reasoning for accurate and interpretable program analysis.

## 5.4 GENERALIZATION TO UNSEEN HARDWARE PLATFORMS

We further evaluate the generalization capability of LLMTuner and prior cost models, which are pretrained on TenSet and directly tested on the dataset collected from unseen hardware without fine-tuning.

**LLMTuner exhibits significant generalization improvement on unseen hardware compared with other cost models.** As shown in Table 3, LLMTuner achieves the highest scores across all evaluated unseen hardware platforms. However, traditional cost models pretrained on TenSet suffer a significant drop in performance. For example, on the NVIDIA A100, LLMTuner achieves Top-1 score of 0.812 and a Top-5 score of 0.902, representing improvements of 30.6% and 22.9% over the results of the traditional cost model, respectively. This strong generalization stems from LLM-Tuner's use of large-scale pretraining and intermediate representations of tensor program execution, which capture complex cross-platform behaviors. In contrast, prior cost models, which rely on hand-crafted features and small models, fail to adapt to new architectures and thus suffer significant performance drops.

## 5.5 ABLATION STUDY

**Ablation of the Two-Stage Framework: Fine-selection stage significantly boosts selection accuracy.** As shown in Table 4, using only the coarse-filtering stage (Stage 1) yields moderate Top-1 score (0.765 on GPU and 0.604 on CPU). When the fine-selection stage is added, Top-1 accuracy increases markedly to 0.919 on GPU (15.4% improvement) and 0.924 on CPU (32.0% improvement). This clearly demonstrates the critical contribution of the fine-selection stage to overall selection accuracy, validating the effectiveness of our reasoning design in accurately distinguishing high-performance programs.

| Hardware | S1 | S2 | Top 1 | Top 5 | Top 10 |
|---|---|---|---|---|---|
| GPU | ✓ |  | 0.765 | 0.911 | 0.946 |
|  | ✓ | ✓ | **0.919** 15.4% ↑ | **0.976** 6.5% ↑ | **0.981** 3.5% ↑ |
| CPU | ✓ |  | 0.604 | 0.914 | 0.937 |
|  | ✓ | ✓ | **0.924** 32.0% ↑ | **0.973** 5.9% ↑ | **0.978** 4.1% ↑ |

Table 4: Ablation Study of the Coarse-Filtering Stage (S1) and the Fine-Selection Stage (S2) on GPU (Nvidia K80) and CPU (Intel Platinum 8276).

|  | K=10 | K=16 | K=32 | K=64 |
|---|---|---|---|---|
| **Top 1** | 0.892 | 0.904 | 0.919 | 0.919 |
| **Top 5** | 0.921 | 0.957 | 0.976 | 0.976 |

Table 5: Impact of the Number of Candidates ($K$) Selected by the Coarse Filtering Stage on Final Performance.

**Ablation on Candidate Set Size: A moderate set size strikes a balance between accuracy and efficiency.** Table 5 examines the effect of varying the number of candidates ($K$) passed from the coarse filtering stage to the fine selection stage. Increasing $K$ improves both Top-1 and Top-5 scores—for instance, Top-1 score rises from 0.892 at $K = 10$ to 0.919 at $K = 32$, with no further gain observed at $K = 64$. However, larger $K$ values also result in increased inference time during the reasoning stage. This indicates that setting $K = 32$ achieves a balanced trade-off between accuracy and efficiency. Overall, this highlights the importance of carefully tuning the candidate set size to optimize resource utilization without sacrificing model performance.

**Ablation of the Reasoning Model: LLMTuner-Reasoner enables accurate inference with lower time.** Table 6 compares the performance of our distilled LLMTuner-Reasoner with DeepSeek-R1 and the untrained Qwen2.5-7B-Instruct baseline. The distilled model achieves Top-1/Top-5 score of 0.919/0.976, closely matching DeepSeek-R1 and significantly outperforming the baseline. Importantly, inference time is drastically reduced from over 30 hours with DeepSeek-R1 to just 1 hour with our distilled model. This

| LLMTuner-Reasoner Model | Top 1 | Top 5 | Time |
|---|---|---|---|
| LLMTuner-Classifier |  |  |  |
| ↪ + Qwen2.5-7B-Instruct | 0.677 | 0.857 | ∼1h |
| ↪ + DeepSeek-R1 671B | **0.924** | **0.976** | ∼30h |
| ↪ + Finetuned Llama3-8B-Instruct | 0.803 | 0.914 | ∼1h |
| ↪ + Finetuned Qwen2.5-1.5B-Instruct | 0.718 | 0.863 | ∼0.6h |
| ↪ + Finetuned Qwen2.5-7B-Instruct | 0.919 | **0.976** | ∼1h |

Table 6: Ablation Study of Different Reasoning Models for Fine-Selection Stage (on K80 dataset).

demonstrates that distillation preserves the reasoning ability while greatly improving efficiency, underscoring the value of task-specific training in achieving accurate and efficient fine selection.

## 6 CONCLUSION

In this paper, we present LLMTuner, a framework that empowers LLMs to perform program performance estimation through a two-stage pipeline for cross-hardware tensor program tuning tasks. By directly reasoning about program execution behavior on hardware, LLMTuner overcomes the high development cost and poor generalization of traditional feature-based cost models. Experiments show LLMTuner achieves superior accuracy and hardware adaptability, outperforming SOTA LLMs and prior cost models. When integrated into the TVM, it demonstrates faster convergence and high final performance on various tensor program tuning tasks, proving its practical effectiveness. This work opens new possibilities for using LLMs as general-purpose performance reasoners for inference acceleration on rapidly evolving models and hardware.

ETHICS STATEMENT

We have considered the ethical implications of our work in accordance with the ICLR Code of Ethics. Our study uses publicly available datasets and does not involve human subjects or sensitive personal data. We have taken steps to mitigate potential biases and misuse of our methods. There are no conflicts of interest or other factors affecting research integrity.

REPRODUCIBILITY STATEMENT

We have made efforts to ensure reproducibility. All model details, experimental settings, and data preprocessing steps are described in the main paper Section 4.4 and 5.1, the appendix Section C. Anonymous source code is provided in https://anonymous.4open.science/r/LLMTuner/

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

## A  THE USE OF LARGE LANGUAGE MODELS (LLMs)

**Writing Refinement.** We utilize GPT-5 and DeepSeek to improve the quality of our written content, focusing on accuracy and natural, native-level expression.

**Research Subject.** In this study, LLMs are also treated as the research subject. We investigate their capabilities in program performance analysis. Specifically, we interact with Gemini, GPT, and Claude via their respective APIs. We also use locally deployed open source LLMs such as Qwen and LLaMA.

## B  DETAILS OF NEWLY COLLECTED DATASET

### B.1  OVERVIEW OF HARDWARE PLATFORMS

The original TenSet dataset, introduced in 2021, consists of program–latency pairs collected from six legacy hardware platforms: Intel Xeon E5-2673, Intel Platinum 8272CL, AMD EPYC 7452, ARM Graviton2, NVIDIA Tesla K80, and NVIDIA Tesla T4. While pioneering at the time, TenSet lacks coverage of modern GPU/TPU and CPU architectures. To bridge this gap, we present **TenSetPerf**, a balanced and large-scale performance dataset containing over 13 million program–latency pairs. As shown in Table 7, it encompasses five diverse and contemporary hardware platforms: Intel Platinum 8575C, ARM Cortex-A76, NVIDIA A100, H100, and RTX 4070. Specifically:

- NVIDIA A100 (Ampere) and H100 (Hopper) are widely adopted for large-scale AI training;
- NVIDIA RTX 4070 (Ada Lovelace) is commonly used for desktop and edge AI inference;
- Intel Xeon Platinum 8575C (Ice Lake SP) represents a modern high-end server CPU, equipped with DDR5 ECC and AVX-512;
- ARM Cortex-A76, a typical flagship mobile/edge SoC core, is increasingly relevant for compact AI inference deployments.

Table 7: Overview of five modern hardware platforms used in the newly collected dataset, including their architectural features, application scenarios, and the number of measured program–latency pairs. The dataset ensures diversity across CPUs and GPUs from both data center and edge environments.

| Hardware | Architecture | Compute Specification | Typical Use Case | # Programs |
|---|---|---|---|---|
| Intel Xeon Platinum 8375C | Ice Lake-SP | 40 cores, AVX-512, DDR5 ECC | Datacenter / inference servers | 920k |
| ARM Cortex-A76 | ARMv8.2 | 8 cores, NEON SIMD units | Mobile / edge platforms | 886k |
| NVIDIA A100 GPU (80GB) | Ampere (2020) | Multi-precision Tensor Cores, HBM2e | HPC / LLM training | 4,808k |
| NVIDIA H100 GPU (80GB) | Hopper (2022) | Transformer Engine, HBMe | AI acceleration / LLM training | 4,808k |
| NVIDIA GeForce RTX 4070 | Ada Lovelace (2023) | 12 GB GDDR6X, RT & Tensor Cores | Single-node inference / graphics | 2,382k |

### B.2  DATA COLLECTION PROCESS

We follow a similar data collection process as TenSet, but we extend it to more tasks and more hardware platforms. First, we select network configurations covering both CV and NLP tasks, including

Table 8: Network Configurations. We vary the batch size and input image dimensions (or sequence lengths) of common network architectures to generate diverse computational graphs of neural networks.

| Network | BatchSize | Image Size (or Seq. Length) |
|---------|-----------|------------------------------|
| Resnet50 | {1, 4, 8} | {224, 240, 256} |
| WideResnet | {1, 4, 8} | {224, 240, 256} |
| Mobilenet-V2 | {1, 4, 8} | {224, 240, 256} |
| Deeplab-V3 | {1, 4, 8} | {224, 240, 256} |
| Mobilenet-V3 | {1, 4, 8} | {224, 240, 256} |
| ResNeXt-50 | {1, 4, 8} | {224, 240, 256} |
| Resnet-3d-18 | {1, 4, 8} | {224, 240, 256} |
| Densenet-121 | {1, 2} | {224, 240, 256} |
| DCGAN | {64, 128} | {64} |
| Inception-V3 | {1, 4, 8} | {299} |
| ViT | {1, 4, 8} | {224} |
| DeTR | {1, 4, 8} | {256} |
| BERT-B | {1, 4} | {128} |
| GPT-2 | {1, 4} | {128} |
| Llama | {1, 4} | {128} |
| OPT | {1, 4} | {128} |

models from PyTorch and Huggingface model zoos, with varying batch sizes and input shapes, as shown in Table 8. For each network, we apply a graph partitioner from Ansor to extract subgraphs, each centered around a heavy tensor operator (e.g., conv2d, matmul). For every subgraph on each of the 5 hardware platforms, we sample up to 4,000 programs from its scheduling search space. Each program is compiled and measured 10 times under warm-up and repeated settings, and the full timing results are recorded in JSON format. After filtering invalid records and deduplication, the new dataset includes over 13 million program–latency pairs.

## C  DETAILS OF DATA PREPARATION AND TRAINING

Anonymous source code is provided in https://anonymous.4open.science/r/LLMTuner/

### C.1  DEFINITION OF TOP K

The top-$k$ score to evaluate the prediction accuracy is defined as:

$$\text{Top-}k = \frac{\sum_m \sum_s \min(T_{m,s}) \times w_{m,s}}{\sum_m \sum_s \min(T_{m,s,i}) \times w_{m,s}}, \quad 1 \le i \le k \tag{5}$$

where $\min(T_{m,s})$ denotes the minimum latency among all tensor programs of subgraph $s$ in model $m$. $w_{m,s}$ is the number of times subgraph $s$ appears in model $m$. $T_{m,s,i}$ denotes the latency of the $i$-th highest-scoring tensor program, as predicted by the cost model, for subgraph $s$ in model $m$.

### C.2  TRAINING OF LLMTUNER-CLASSIFIER

We adopt Qwen2.5-0.5B-Instruct as the base model for LLMTuner-Classifier. This lightweight language model supports long-context inputs (max to 32,768 tokens) and is extended with a shallow classification head for binary classification(as the definition in `AutoModelForSequenceClassification()`). The classifier is trained on labeled data from the TenSet dataset, where each candidate tensor program is annotated as either "good" or "bad" based on its measured runtime. Specifically, within each subgraph, we empirically label the top $k$ candidates with the lowest runtimes as "good", where $k$ is defined as the larger of 32 or 10% of all valid programs in the subgraph, while all remaining candidates are labeled as "bad".

To address class imbalance between "good" and "bad" samples, we adopt a two-fold strategy:

- We apply a controlled downsampling approach to limit the dominance of the majority class. Specifically, for each subgraph, we retain all "good" samples and at most twice as many "bad" samples, i.e.,

$$|\mathcal{D}_{\text{bad}}| = \min\left(|\mathcal{D}_{\text{bad}}|, 2 \cdot |\mathcal{D}_{\text{good}}|\right). \tag{6}$$

This prevents the classifier from being overwhelmed by excessive bad samples and encourages more balanced learning.

- We employ a class-weighted MSE loss to account for any remaining imbalance. The weight for each sample is computed based on the inverse frequency of its class in the training set:

$$w_i = N/N_{y_i}, \quad y_i \in \{0, 1\}, \tag{7}$$

where $N$ is the total number of training samples and $N_{y_i}$ is the number of samples in class $y_i$. The final loss is defined as:

$$\mathcal{L}_{\text{cls}} = \frac{1}{N} \sum_{i=1}^{N} w_i \cdot (y_i - \hat{y}_i)^2 \tag{8}$$

## C.3 TRAINING OF LLMTUNER-REASONER

To enable efficient and scalable structured reasoning, we adopt a knowledge distillation approach to train the LLMTuner-Reasoner. We first collect reasoning data from DeepSeek-R1, then fine-tune the base model with these reasoning data.

**Reasoning Data Curation.** We first employ a powerful LLM, DeepSeek-R1, as the teacher model to generate high-quality reasoning responses and program–hardware alignment scores for approximately 100K prompt–response pairs. These prompts are sampled from the TenSet dataset across six representative hardware platforms.

To ensure data quality and diversity, we apply two strategies: (1) only programs labeled as `good` in the dataset are selected for distillation; (2) for a subset of examples, **real latency measurements are provided as additional input** to DeepSeek-R1 to guide its reasoning. Meanwhile, as shown in Listing 1, carefully designed instructions are incorporated into the prompt format to encourage the model to learn reasoning patterns that generalize to scenarios without access to latency, promoting robustness. All generated responses undergo manual review and filtering to eliminate low-quality or ambiguous cases.

```
'''
**Hint:**
hint_score ={normal_time} is a normalized performance score of the giving TVM program in range
    [0.0, 1.0], where values closer to 1.0 indicate better performance.
If hint_score is provided, use it as an auxiliary signal to guide scoring decisions (e.g.,
    higher hint_score may imply better utilization and lower bottlenecks).
If hint_score is None, score solely based on schedule features – your output should remain
    valid and reasonable.
Never include hint_score in the final reasoning output.
'''
```

Listing 1: Example of the Hint Used to Guide the Reasoning.

**Supervised Fine-Tuning.** The curated prompt–response pairs are then used to fine-tune a smaller, open-source student model, `Qwen2.5-7B-Instruct`, using parameter-efficient LoRA-based supervised fine-tuning (SFT). This allows the student model to effectively inherit the structured reasoning capabilities of the large teacher model while remaining lightweight and deployment-friendly.

Let $\mathcal{D} = \{(x_i, y_i)\}_{i=1}^{N}$ denote the set of curated prompt–response pairs, where $x_i$ is a reasoning prompt and $y_i$ is the corresponding teacher-generated response. The SFT objective minimizes the cross-entropy loss between the student model's output $P_\theta(y_i \mid x_i)$ and the teacher-provided target:

$$\mathcal{L}_{\text{SFT}}(\theta) = -\sum_{i=1}^{N} \log P_\theta(y_i \mid x_i)$$

To enhance training efficiency, we incorporate LoRA, which injects trainable low-rank adapters into the attention and feedforward layers, reducing memory and compute overhead.

## C.4 TRAINING OF REGRESSION MODEL

We train a regression model to map the program–hardware alignment scores to predicted runtime values. The training data consists of alignment scores generated by DeepSeek R1, paired with ground-truth runtime measurements from the TenSet dataset. All runtime values are linearly normalized to the range $[0, 1]$ for stable training.

To address data imbalance across different runtime regions, we employ a binning-based oversampling strategy. Specifically, the normalized runtime values are divided into $N = 10$ equal-width bins:

$$B_j = \left\{ i \mid r_i \in \left[ \frac{j-1}{N}, \frac{j}{N} \right) \right\}, \quad j = 1, 2, \ldots, N. \tag{9}$$

where $B_j$ denotes the set of sample indices whose normalized runtime $r_i$ falls into the $j$-th bin. We then oversample each bin to match the size of the largest bin, ensuring uniform coverage across the runtime spectrum and preventing bias toward frequently occurring ranges.

The regression model is implemented as a three-layer MLP and trained with a combined loss function that balances ranking loss and mean squared error (MSE) loss:

$$L = \lambda \cdot \text{ranking\_loss} + (1 - \lambda) \cdot \text{MSE\_loss}, \tag{10}$$

where $\lambda$ is a weighting coefficient. This procedure enables the model to learn a fine-grained and generalizable mapping from semantic performance assessments to normalized execution times across diverse hardware platforms.

## D AN EXAMPLE INPUT TO LLMTUNER-CLASSIFIER

This section presents an example input to the `LLMTuner-Classifier`. The input is formulated as a structured prompt containing both program and hardware specifications. The example in Listing 2 demonstrates how these components are combined into a single prompt for inference.

## E AN EXAMPLE INPUT TO LLMTUNER-REASONER

To enable structured performance reasoning, LLMTuner-Reasoner receives as input a task description that integrates program details, hardware specifications, and key evaluation metrics provided in JSON format. The prompt instructs the model to generate detailed and interpretable analyses of the program's performance characteristics, with the output required in JSON format. In Listing 3, we present an example input used for reasoning. In Listing 4, we present the key evaluation metrics in JSON format.

```python
def prompt_for_classification(program, hardwareinfo):
    prompts = f"""You are an expert in compiler optimization and hardware performance analysis.

Your task is to evaluate the following tensor program on the given hardware and judge whether
    its performance is acceptable.

# Tensor Program:
{program}

# Hardware Specifications:
{hardwareinfo}

# Evaluation Criteria:
- Consider computation efficiency, memory access, and parallelism.
- Base your judgment on general performance best practices.

# Output:
Respond with one word only: "good" or "bad".
"""
    return prompts

program = """
blockIdx.x ax0.0@ax1.0@ax2.0@ax3.0@ax4.0@ (0,56)
  vthread ax0.1@ax1.1@ax2.1@ax3.1@ax4.1@ (0,2)
    threadIdx.x ax0.2@ax1.2@ax2.2@ax3.2@ax4.2@ (0,448)
      Conv3dOutput auto_unroll: 64
      for rh.0 (0,3)
        for rc.0 (0,128)
          threadIdx.x ax0@ax1@ax2@ax3@ax4@.0.1 (0,448)
            vectorize ax0@ax1@ax2@ax3@ax4@.1 (0,32)
              placeholder.shared = ...
          threadIdx.x ax0@ax1@ax2@ax3@ax4@.0.1 (0,448)
            vectorize ax0@ax1@ax2@ax3@ax4@.1 (0,21)
              PaddedInput.shared = ...
          for rd.1 (0,3)
            for nn.3 (0,4)
              for rw.2 (0,3)
                for rc.2 (0,4)
                  Conv3dOutput = ...
      for ax0.3 (0,4)
        T_relu = ...
"""

hardware_info = """
{
    "GPU": "NVIDIA K80",
    "SMs": 13,
    "MaxThreadsPerBlock": 1024,
    "MaxSharedMemoryPerBlockKB": 48,
    "WarpSize": 32,
    "BaseClockSpeedMHz": 560,
    "BoostClockSpeedMHz": 875,
    "MemoryBandwidthGBs": 240,
    "PeakFP32TFLOPs": 8.7
}
"""
```

Listing 2: An Example Input to LLMTuner-Classifier.

```python
def prompt_for_reasoner(program, hardwareinfo, normal_time=None):

    key_eval_metric = json.dumps(EVAL_METRIC, indent=2, ensure_ascii=False)

    prompt = f"""Your task is to analyze a given TVM schedule and target hardware information
        to generate a detailed textual feature report. This report will be used as input for a
        downstream cost model.

Focus on the interaction between the schedule's characteristics and the hardware's
    capabilities and limitations. Please populate the following JSON template with
    quantitative assessments where specified, alongside textual explanations.

**TVM Programs:**
```
{program}
```

**Hardware Information (JSON):**
```json
{hardwareinfo}
```

**JSON Template to Populate:**
```json
{key_eval_metric}
```

**General Scoring Guidance:**
- Scores ending with '_score' are typically in the range 0.0 to 1.0 unless otherwise specified
    (e.g., -1.0 to 1.0, or integer scales like 1-5). Generally, 1.0 or the higher end of a
    positive scale indicates an optimal or desirable state, while for risk/severity, a higher
    score indicates greater concern.
- Integer levels (e.g., for `register_pressure_and_spilling_risk_level`) should strictly
    follow the defined scale in their description.
- For lists like `bottlenecks`, populate them according to the description.

Focus on identifying the most critical performance aspects and provide your best quantitative
    judgment directly in the score/level fields, supported by brief reasoning in the
    corresponding text fields.
Ensure your output is a single, valid JSON object matching the structure of the 'JSON Template
    to Populate', with all fields filled.

**Important Instructions:**
1. Please strictly follow the structure of the 'JSON Template to Populate' provided above.
2. Your output **must be and can only be** a valid JSON object.
3. Do **not** add any introductory text, explanations, comments, or markdown tags (such as ```
    json ... ```) before or after the JSON object.
4. Make sure the JSON object starts with `{{` and ends with `}}`.
"""
    return prompt
```

Listing 3: An Example Input to LLMTuner-Reasoner.

```
EVAL_METRIC = {
  "parallelism_assessment": {
    "description": "Assessment of GPU-level parallelism efficiency across SMs, warps, and
        threads.",
    "sm_utilization_score": {
      "description": "Measures how effectively the grid of thread blocks covers Streaming
          Multiprocessors (SMs), ensuring no SM starvation or oversubscription. \
Score range: 0.0  1.0 (1.0 = perfect SM coverage).",
      "score": 0.0,
      "reasoning_text": ""
    },
    "warp_execution_efficiency_score": {
      "description": "Evaluates warp efficiency, including whether warps are fully populated and
          the degree of divergence in control flow execution. \
Score range: 0.0  1.0 (1.0 = no divergence, fully efficient).",
      "score": 0.0,
      "reasoning_text": ""
    },
    "occupancy_score": {
      ...
    },

  "memory_assessment": {
    ...
  },
  "computation_assessment": {
    ...
  }
}
```

Listing 4: An Example of the Evaluation Metrics in JSON Format.

---

**Algorithm 1 Iterative Tensor Program Tuning Workflow**

---

**Input**: Subgraph set $\mathcal{G}$, target hardware $h$, number of candidates per round $n$, number of rounds $R$, predictive cost model $\hat{T}$

**Output**: Best program $p^*$

1: $\mathcal{P} \leftarrow \emptyset$          // Initialize candidate program set
2: **for** each subgraph $g$ in $\mathcal{G}$ **do**
3:     $\mathcal{P}_g \leftarrow$ ConstructSearchSpace($g$)
4:     $\mathcal{P} \leftarrow \mathcal{P} \cup \mathcal{P}_g$
5: **end for**
6: $S \leftarrow \emptyset$          // Initialize evaluation history
7: **for** round $r = 1$ to $R$ **do**
8:     $\mathcal{C}_r \leftarrow$ SearchAlgorithm($\mathcal{P}, S, n$)        // Generate $n$ candidates using feedback
9:     **for** each $p$ in $\mathcal{C}_r$ **do**
10:       **if** $\hat{T}$ is available **then**
11:         $score \leftarrow \hat{T}(p, h)$
12:       **else**
13:         $score \leftarrow T(p, h)$
14:       **end if**
15:       $S \leftarrow S \cup \{(p, score)\}$
16:     **end for**
17: **end for**
18: $p^* \leftarrow \arg\min_{(p, score) \in S} score$
19: **return** $p^*$

---

