# OpenReview forum: "One for all: Zero-Shot Cross-Hardware Performance Modeling with LLMs for Tensor Program Tuning"
_ICLR.cc/2026/Conference — Submitted to ICLR 2026_

### Official Review · Reviewer_DUzA · 2025-10-24

**Soundness:** 2
**Presentation:** 3
**Contribution:** 1
**Rating:** 2
**Confidence:** 4

**Summary:**

The paper proposes using large language models (LLMs) to predict and select high-performance kernels from a set of candidates, a process known as kernel tuning. The proposed system, LLMTuner, employs an LLM-based classifier for coarse-grained selection of potentially optimal kernels, followed by a fine-grained selection stage. In the second stage, a fine-tuned LLM predicts execution behavior and selects the best kernel based on its inference. Experiments show that LLMTuner improves estimation accuracy by up to 64.8% compared with general-purpose LLMs and other cost models.

**Strengths:**

1. The paper proposes a framework for predicting execution behavior and performing kernel tuning.
2. A coarse-to-fine framework improves both efficiency and accuracy.
3. Experimental results demonstrate improved accuracy compared with general-purpose LLMs and other cost models.

**Weaknesses:**

1. Weak motivation
2. Weak evaluation

**Questions:**

Thank you to the authors for submitting to ICLR 2026.

I appreciate the idea of leveraging LLMs to analyze kernel programs and demonstrate their ability to recognize efficient kernels. However, as a paper introducing a cost model for kernel tuning, this work has relatively weak motivation and evaluation.

**1. Weak motivation**

During the era of CNN optimization, systems such as TVM and its schedulers (AutoTVM/Ansor) were widely used for end-to-end compilation. At that time, the search space for even a single convolution kernel was enormous, making exhaustive benchmarking infeasible. Thus, AutoTVM proposed using cost models (e.g., XGBoost, MLP) to guide the search for high-performance schedules. These models were crucial for accelerating the tuning process.

However, with the evolution of deep learning and the increasing importance of LLMs, most kernels are now produced by compilers such as Triton, template libraries such as CUTLASS or Composable Kernels, or vendor libraries such as cuBLAS. In frameworks like Triton, the search space is much smaller (typically 10 to 100 candidates), making it feasible to compile and benchmark all candidates directly on the target hardware. Therefore, the motivation for using a cost model like LLMTuner in this context needs to be strengthened.

**2. Weak evaluation**

The paper builds on the TenSet dataset, which is relatively outdated, as noted in Appendix B. The authors extend it to TenSetPerf by adding more hardware and workloads. However, the programs are collected using Ansor, which, to my knowledge, does not support Tensor Cores, the key computational units in modern GPUs, and provides limited support for dynamic shapes, which are essential for LLM workloads (e.g., prefill operations).

It would strengthen the paper if the authors could evaluate their approach on real workloads (e.g., Llama-3, or ResNet50) and compare the best kernels found by LLMTuner with state-of-the-art implementations from Triton, CUTLASS, cuDNN or cuBLAS.

That said, I believe the authors’ exploration has potential value. A promising future direction might be to use LLMs not just for tuning but also for generating efficient kernel code.

---

> ### Author Response · Authors · 2025-11-24
> **Response for Reviewer DUzA (Part #1)**
>
> > W1/Q1: Weak motivation
>
> We clarify that auto-tuning and cost models remain essential in 2025, especially beyond template-based compilers such as Triton/CUTLASS. Recent research continues to expand TVM-style auto-scheduling to new architectures and workloads.
>
> For example, MetaSchedule has been extended to optimize tensor programs on the RISC-V Vector ISA, achieving significant speedups over compiler autovectorization [1]. The ML compiler community (e.g., C4ML at CGO 2025) continues to emphasize automatic scheduling and cost-model design as core research problems [2]. Moreover, systems research shows that fused operators and irregular kernels in modern LLM inference still rely on auto-tuned schedules rather than static templates [3].
>
> These developments demonstrate that cost models remain widely needed, particularly for expressive search spaces, emerging ISAs, and model-specific fused kernels where template-based approaches cannot cover the full optimization space. LLMTuner further strengthens motivation by providing a generalizable and hardware-adaptive cost model, reducing per-device retraining and enabling portability across diverse hardware platforms.
>
> [1]Tensor Program Optimization for the RISC-V Vector Extension Using Probabilistic Programs.(arXiv 2025)
>
> [2]C4ML: Compilers for Machine Learning Workshop.(CGO 2025)
>
> [3]Accelerating LLMs Using an Efficient GEMM Library and Target-Aware Optimizations on Real-World PIM Devices.(CGO 2025)

---

> ### Author Response · Authors · 2025-11-24
> **Response for Reviewer DUzA (Part #2)**
>
> > W2/Q2:Weak evaluation
>
> We thank the reviewer for the comment. While TenSet is an older benchmark, it remains the only large-scale public benchmark for cost-model research and continues to be adopted in recent works such as TLP (ASPLOS’23) and Pruner (ASPLOS’25). We further extend it to TenSetPerf by incorporating new GPU generations, CPUs, and additional workloads, substantially broadening its coverage.
>
> We choose Ansor over MetaSchedule for two reasons:(1)Cost models are pre-trained on TenSet, providing a consistent evaluation framework with our extended dataset. (2)Our goal is to evaluate a generalizable performance model under a large and diverse search space. Ansor generates much richer schedule variance compared to MetaSchedule’s rule-based sketches, making it more suitable for training and stress-testing a hardware-agnostic cost model.
>
> Figure 4 presents LLMTuner’s performance on real workloads, including comparisons with relevant prior work. Following the reviewer’s suggestion, we also evaluate LLMTuner on ResNet-50 and Bert-base kernels, comparing against Triton(torch.compile) and cuDNN/cuBLAS(torch eager). We using FP32 precision for all evaluations, since LLMTuner are training on TenSet dataset(searched by Ansor using FP32). Here are the results：
>
> | **Workload**  | **Input Shape**  | **cuDNN/cuBLAS** | **Triton** | **LLMTuner** |
> | ------------- | ---------------- | ---------------- | ---------- | ------------ |
> | **ResNet50**  | (1, 3, 224, 224) | 4.07ms           | 1.55ms     | 1.42ms       |
> | **BERT-base** | (1, 128)         | 4.58ms           | 3.06ms     | 3.65ms       |
>
> Results show that LLMTuner consistently outperforms vendor libraries (cuDNN/cuBLAS) and discovers FP32 kernels competitive with or even superior to Triton's compiler-generated code (e.g., 1.09x speedup on ResNet50), demonstrating its capability to find optimal schedules effectively.
>
> Our long-term goal aligns precisely with the reviewer’s vision: using LLMs not only for tuning but also for generating high-performance kernel code. LLMTuner represents the first step in this direction by introducing a behavior-centric performance modeling paradigm, and in the future, enabling an “LLM-as-generator, LLM-as-performer, closed-loop self-improving system”.
>
> In summary, although TenSet is an older benchmark, its continued use in recent literature, our extended TenSetPerf dataset, and additional real-world workload evaluations collectively provide solid empirical support for LLMTuner. We appreciate the suggestion and emphasize that our performance model is explicitly designed to support the next generation of LLM-driven kernel generation systems.

---

> ### Comment · Reviewer_DUzA · 2025-11-24
>
> Thank you for your detailed response addressing my initial concerns. I appreciate the clarification and the additional experimental results. However, I still find the motivation and evaluation to be relatively weak for a cost-model paper in the context of modern compiler and hardware trends.
>
> ### 1. Weak Motivation (W1/Q1)
>
> While the authors argue that TVM-style auto-scheduling and cost models remain relevant in exploratory research (RISC-V, fused kernels), the core issue raised in my original review has not been resolved:
>
> * **Modern Context:** The response does not adequately address the ubiquity of Tensor Cores (key computational units in modern GPUs) and the need for dynamic shape support (crucial for LLM prefill operations), which are generally not supported by the Ansor/TenSet framework. If the target application space is modern LLMs, these features are non-negotiable.
> * **Search Space:** For the majority of high-volume, performance-critical kernels (e.g., GEMM), the search space for state-of-the-art systems like Triton and CUTLASS is already small and manageable (10-100 candidates). The motivation for a complex LLM-based cost model to replace a simple compile-and-benchmark approach in this dominant regime remains weak.
> * **Dataset Bias:** Stating that older benchmarks like TenSet are "adopted in recent works" does not alleviate the concern; it merely highlights a potential weakness in the evaluation of those papers. The continued reliance on a dataset inherently tied to the Ansor scheduler raises questions about the model's performance on schedules generated by state-of-the-art tools (Triton/MetaSchedule/CUTLASS). Used by recent publications is not a sufficient answer to the concern that the dataset is potentially biased toward a specific, older scheduler.
>
> ### 2. Weak Evaluation (W2/Q2)
>
> The new results comparing LLMTuner against cuDNN/cuBLAS and Triton on ResNet-50 and BERT-base are a good step, but they raise more questions:
>
> * **Kernel vs. System Performance:** The comparison on ResNet50 (4.07ms vs. 1.42ms, a 2.87x speedup) is surprising, as cuDNN/cuBLAS are highly optimized kernel libraries. This result strongly suggests that the comparison is capturing graph-level optimizations (like whether use cuda graph, whether perform some graph-level optimization like fusion) rather than a pure kernel-to-kernel comparison. For a cost-model paper, a kernel-level performance comparison is essential to demonstrate the model's core effectiveness. Is there any other orthogonal factor (like graph-level optimization, using CUDA graph) involved?
> * **BERT-base Performance:** The execution time for BERT-base shows LLMTuner is slower than Triton (3.65ms vs. 3.06ms). This is especially concerning since the cost model in Triton's typical usage is solved in a super simple, highly effective way: run the few candidate kernels and pick the best one. This result contradicts the claimed efficacy of the LLMTuner cost model in a real-world, modern context.
> * **Precision:** Furthermore, the cost model is evaluated using FP32. Modern LLM workloads are almost exclusively run in lower precision (FP16/BF16) to fully utilize Tensor Cores.
>
> In summary, the provided context and additional results do not sufficiently strengthen the paper's core claims regarding motivation and evaluation within the landscape of modern kernel tuning. While I appreciate the exploration of LLMs for performance modeling, the paper needs to demonstrate its value proposition by comparing its cost model against the state-of-the-art search/compilation strategies used for performance-critical kernels today.

---

> > ### Author Response · Authors · 2025-11-28
> > **Response for Reviewer DUzA (Part #1)**
> >
> > We sincerely thank the reviewer for the constructive criticism regarding the relevance of our evaluation platform and the motivation in the era of Triton/Tensor Cores. We value your expertise and have taken your feedback to heart.
> >
> > To address your core concerns regarding outdated baselines (Ansor/TenSet), we have conducted substantial new experiments by integrating LLMTuner into **MetaSchedule** (the next-generation TVM scheduler supporting FP16 and Tensor Cores) to demonstrate the robustness and modernity of our approach, and we will add these results into the revision.
> >
> > > W1: Weak motivation
> >
> > > W1.1 Modern Context
> >
> > LLMTuner is applicable in modern compilation environments (e.g., Tensor Cores), we integrated LLMTuner into MetaSchedule, TVM's newest auto-scheduler. MetaSchedule offers advanced search capabilities and native support for FP16 precision and Tensor Core intrinsics.
> >
> > LLMTuner can also support dynamic shapes and operators. By reasoning over execution behavior, it generalizes to unseen shapes and operator patterns, providing a significant advantage over traditional cost models that not support for new inputs.(As shown in Tab 3 of the paper)
> >
> > Intergrated to MetaSchedule, we evaluated LLMTuner along two dimensions to demonstrate its value in the modern context.
> >
> > **1. Generalization Across Operator Types and Unseen Shapes**
> >
> > We first collected a small dataset from MetaSchedule tuning logs in FP16/Tensor Core mode on an NVIDIA A100 GPU, comprising 70 convolution operators and 50 matmul operators with different input shapes. To assess LLMTuner’s ability to generalize across operator types and input shapes, we selected 10 matmul and conv operators with different input shapes as the test set. LLMTuner was fine-tuned only on conv-related operators, and we evaluated prediction accuracy on:
> >
> > - conv test set (same operator type, unseen shapes)
> > - matmul test set (entirely unseen operator type)
> >
> > | **Training: Conv Only** | **Top-1** | **Top-5** |
> > | ----------------------- | --------- | --------- |
> > | **Test: Conv**          | 100.0%    | 100.0%    |
> > | **Test: MatMul**        | 99.4%     | 100.0%    |
> >
> > The high prediction accuracy on the unseen MatMul operators confirms that LLMTuner captures shape-agnostic and operator-agnostic execution behaviors that are crucial for modern dynamic workloads. This capability directly addresses the reviewer's concern regarding modern context.
> >
> > **2. Search Efficiency (Online Tuning)**
> >
> > We integrated LLMTuner as the cost model within the MetaSchedule search workflow and compared its search efficiency against MetaSchedule’s default cost model.
> >
> > We targeted a performance-critical LLM kernel: a `batch_gemm` operator with inputs of shape (bs, M, K) and (bs, K, N), where bs=4, M=4096, K=4096, N=4096, in FP16/Tensor Core mode on A100. We compared the best latency found within 1000 trials.
> >
> > |              | **Best Latency** |
> > | ------------ | ---------------- |
> > | MetaSchedule | 3.73 ms          |
> > | LLMTuner     | **3.65 ms**      |
> >
> > The LLMTuner-guided search found a better kernel faster than the default MetaSchedule cost model. This result proves that LLMTuner not only integrates into the modern compiler stack (Tensor Core/FP16) but also  accelerates the tuning process for performance-critical LLM kernels, providing direct efficiency value.
> >
> > > W1.2 Search Space
> >
> > We agree that for isolated, highly optimized kernels such as GEMM, the search space in Triton or CUTLASS is sufficiently small to allow exhaustive benchmarking. This is because their tuning spaces are defined by manually engineered templates and a fixed set of hand-designed parameters, which inherently limit extensibility and constrain the overall search space. However, LLMTuner is not designed for these narrow operator-specific cases.
> >
> > Our contribution focuses on network-level, end-to-end optimization in modern compilers, where subgraph fusion, scheduling choices, and cross-operator interactions create a large, combinatorial search space that is fundamentally different from single-operator tuning. In such settings, the candidate space is no longer 10–100 but effectively unbounded, making compile-and-benchmark impractical and requiring a strong cost model.
> >
> > Additionally, Triton and CUTLASS are vendor-specific and hardware-restricted, and must be extensively re-engineered for emerging accelerators. LLMTuner, by contrast, generalizes across hardware platforms（6 CPUs and 5 GPUs）, providing predictive guidance on new  devices where no optimized libraries exist.
> >
> > Thus, our method is motivated not by single-operator tuning, but by the need to scale optimization to full-network compiler workloads and diverse hardware targets, where an effective, generalizable cost model becomes essential.

---

> > ### Author Response · Authors · 2025-11-28
> > **Response for Reviewer DUzA (Part #2)**
> >
> > > W1.3 Dataset Bias
> >
> > The reviewer concerned that TenSet is biased toward Ansor. By validating LLMTuner on the MetaSchedule search space, which generates significantly different schedule structures compared to Ansor, we show that LLMTuner captures fundamental execution behaviors rather than overfitting to a specific scheduler's patterns.
> >
> > > W2: Weak evaluation
> >
> > > W2.1 Kernel vs. System Performance
> >
> > We confirm that the 2.87x speedup observed on ResNet-50 stems from compiler-driven, end-to-end graph-level optimizations, primarily subgraph fusion, rather than solely isolated kernel replacement. We also clarify that neither the cuDNN/cuBLAS baseline nor the LLMTuner-optimized TVM pipeline explicitly utilized CUDA Graph in our measurements.
> >
> > The focus on subgraph fusion is intentional and crucial to the motivation of LLMTuner:
> >
> > - Network-Level Optimization: LLMTuner is designed as a network-level, end-to-end compiler optimization tool, not merely a single-operator micro-tuning cost model. Its core utility is to guide the compiler (Ansor/MetaSchedule) toward optimal fusion and scheduling decisions for complex, non-template-based operator sequences.
> > - Leverage in Non-Vendor Kernels: Vendor libraries like cuDNN/cuBLAS operate at the single-operator level and cannot inherently exploit cross-operator fusion opportunities. Our result demonstrates LLMTuner's intended strength: delivering substantial performance gains in complex, fused workloads where traditional kernel libraries have no direct leverage.
> >
> > > W2.2 BERT-base Performance
> >
> > We acknowledge that Triton outperforms our auto-generated kernels on this specific workload (3.06 ms vs. 3.65 ms). This is expected: Triton relies on expert-crafted, hand-optimized templates for common operators, while LLMTuner operates in a fully automated manner. Despite the lack of manual engineering, LLMTuner’s performance is only 0.6 ms slower than Triton’s expert kernels, while additionally offering cross-hardware generalization and requiring no operator-specific templates or tuning rules. This highlights LLMTuner’s practicality for long-tail operators, new architectures, and emerging hardware platforms where expert templates are unavailable.
> >
> > > W2.3 Precision (FP32 vs. FP16)
> >
> > We addressed this limitation directly with our new MetaSchedule experiments.
> >
> >
> >
> > In summary, by integrating LLMTuner into MetaSchedule with FP16/Tensor Core support, we demonstrate strong generalization, modern relevance, and improved search efficiency on real LLM kernels. While Triton benefits from expert-crafted templates, LLMTuner provides a fully automated, hardware-generalizable, and template-free solution. Importantly, we clarify that LLMTuner is designed for network-level, end-to-end optimization, not merely single-operator tuning. We hope these new results effectively address the reviewer’s concerns.

---

### Official Review · Reviewer_wniB · 2025-10-30

**Soundness:** 3
**Presentation:** 3
**Contribution:** 3
**Rating:** 6
**Confidence:** 4

**Summary:**

This paper proposes LLMTuner, a novel framework that leverages large language models to analyze tensor program execution behaviors and estimate performance across diverse hardware. This paper introduces a coarse-to-fine design in which a lightweight LLM-based classifier filters suboptimal programs and a finetuned LLM infers execution behavior scores to predict latency. This paper demonstrates strong empirical results compared to existing cost models while maintaining over faster inference speed than DeepSeek R1.

**Strengths:**

1. This paper proposes a novel cross-hardware cost modeling paradigm that leverages large language models to analyze unified program execution behaviors, effectively overcoming the limitations of traditional hardware-specific cost models.

2. This paper designs a well-structured coarse-to-fine framework that fine-tunes LLMs for accurate and efficient tensor program performance estimation across diverse hardware platforms.

3. This paper demonstrates strong empirical results, showing substantial accuracy gains and tuning efficiency improvements across 11 hardware types, while maintaining low inference cost with lightweight finetuned models.

**Weaknesses:**

1. This paper claims that "our approach removes the dependence on hand-crafted features". However, in the methodology, the proposed framework uses LLM to evaluate three core dimensions "computation, memory, parallelism". The two statements appear to be contradictory.

2. In hardware-aware structured reasoning, this paper considers three core dimensions: parallelism, memory, and computation. Why do you ignore communication? Could you please clarify it?

3. There is no ablation study for multi-dimension scores (e.g., parallelism, memory, computation). It would be hard to verify whether these "execution behavior scores" truly capture transferable performance factors.

**Questions:**

Why ignore communication but only consider three dimensions of parallelism, memory, and computation in hardware-aware structured reasoning?

---

> ### Author Response · Authors · 2025-11-24
> **Response for Reviewer wniB**
>
> We sincerely appreciate Reviewer wniB’s acknowledgment of our novel cross-hardware modeling paradigm, well-designed coarse-to-fine LLM framework, and strong, efficient empirical results across diverse hardware, which further reinforces the value of LLMTuner for accurate and generalizable tensor program performance modeling.
>
> Below we address the specific comments:
>
> > W1: “hand-crafted features”  vs. “three dimensions score”
>
> Our claim of “removing dependence on hand-crafted features” refers to removing the large, hardware-specific hand-crafted feature sets required by prior cost models. For example, Ansor[1]/TenSet[2] manually extract 164-dimension features for each program, while TIRAMISU[3] extracts 2534-dimension features, each requiring expert-designed rules tied to specific hardware. Such feature engineering is labor-intensive and must be redesigned for every new hardware, which fundamentally limits transferability.T hese features mainly describe the program itself, ignoring hardware interactions.
>
> In contrast, our three dimensions (parallelism, memory, computation) are not hand-crafted numerical features, but high-level semantic categories that structure LLM reasoning. All quantitative sub-scores under these dimensions are automatically inferred by the trained LLM, without any manually defined formulas or hardware-dependent extraction rules. They capture the execution behavior of programs on hardware.
>
> [1]Ansor: Generating High-Performance tensor programs for deep learning(OSDI 20)
>
> [2]Tenset: A large-scale program performance dataset for learned tensor compilers(NeurIPS 21)
>
> [3]A deep learning based cost model for automatic code optimization(MLSys 21)
>
> > W2/Q1: Why ignore communication, only consider parallelism, memory, computation?
>
> In the tuning scenarios covered by TenSet and TVM, all tensor programs are executed on the same device, and thus there is no inter-device communication (e.g., multi-GPU transfer). Such device-level communication lies outside the scope of program-level performance modeling. Within a single device, the remaining forms of “communication”—such as DRAM–cache transfers, shared-memory traffic, and thread/block synchronization—are inherently expressed through memory behavior (data movement efficiency, locality, bandwidth) and parallelism behavior (synchronization cost, stall patterns). Therefore, communication is not ignored; its performance impact is fully captured within the memory and parallelism dimensions.
>
> > W3: Ablation study for multi-dimension scores
>
> Here is the ablation study of multi-dimension scores. Experiments are done on Tenset K80 dataset. For each ablation configuration, we remove specific dimensions of the multi-dimension scores and retrain the regression model accordingly.
>
> | Parallelism (6-dim) | Memory (6-dim) | Computation (4-dim) | Top-1     | Top-5     |
> | ------------------- | -------------- | ------------------- | --------- | --------- |
> | ✔                   | ✔              | ✔                   | **0.919** | **0.976** |
> | ✘                   | ✔              | ✔                   | 0.821     | 0.923     |
> | ✔                   | ✘              | ✔                   | 0.755     | 0.905     |
> | ✔                   | ✔              | ✘                   | 0.884     | 0.925     |
> | ✔                   | ✘              | ✘                   | 0.793     | 0.851     |
> | ✘                   | ✔              | ✘                   | 0.785     | 0.840     |
> | ✘                   | ✘              | ✔                   | 0.651     | 0.804     |
>
> The results clearly demonstrate that each dimension contributes substantially to the model’s predictive performance. When only one dimension is removed, the impact on Top-1 accuracy is pronounced, whereas the decrease in Top-5 accuracy remains relatively small. In contrast, removing two dimensions leads to a much more severe degradation in both Top-1 and Top-5 performance, indicating that the model heavily depends on the joint contribution of all three dimensions to maintain robust prediction quality.

---

### Official Review · Reviewer_Guhc · 2025-10-31

**Soundness:** 2
**Presentation:** 2
**Contribution:** 2
**Rating:** 4
**Confidence:** 4

**Summary:**

This paper addresses the bottleneck of traditional tensor program cost models—high development costs, poor efficiency, and limited cross-hardware generalization (due to reliance on manual hardware-specific features and extensive profiling)—by proposing LLMTuner, a two-stage coarse-to-fine framework that leverages large language models (LLMs) for cross-hardware tensor program performance estimation.
- Coarse Filtering Stage: A lightweight LLM-based classifier (using Qwen2.5-0.5B-Instruct) rapidly filters out suboptimal tensor programs, reducing computational overhead for subsequent fine selection. The classifier is trained on the TenSet dataset, with programs labeled "good" (top-low latency candidates) or "bad" (remaining candidates).
- Fine Selection Stage: A fine-tuned LLM (Qwen2.5-7B-Instruct via LoRA) performs hardware-aware structured reasoning to infer multi-dimensional execution behavior scores (covering parallelism, memory, and computation metrics). These scores are fed into a 3-layer MLP regression model to predict actual program latency, guiding selection of high-performance candidates.
- Proposes a cross-hardware cost modeling paradigm that uses LLMs to analyze unified program execution behaviors (e.g., SM utilization, cache hit rates) instead of hardware-specific static features, overcoming limitations of traditional models.
- Validates LLMTuner’s effectiveness across 6 CPU and 5 GPU platforms (including legacy hardware like NVIDIA K80 and modern platforms like A100/H100): it improves performance estimation accuracy by up to 64.8% vs. general-purpose LLMs (e.g., Gemini, GPT-4o) and traditional cost models (e.g., Ansor, TenSetMLP), and achieves 49.2% accuracy improvement on unseen hardware.
- Demonstrates practical value for DNN/LLM tuning: LLMTuner discovers 1.47× better program performance with up to 3.27× higher tuning efficiency vs. baselines, and reduces estimation time by over 30× vs. DeepSeek R1 when using fine-tuned lightweight LLMs.

**Strengths:**

-  Addresses a practical issue: Traditional tensor program cost models have poor cross-hardware generalization; the work’s motivation is relevant for LLM deployment across diverse hardware.
- Logical framework design: The coarse-to-fine (lightweight LLM classifier + fine-tuned LLM for execution behavior reasoning) structure balances efficiency and accuracy.
- Broad experimental scope: Covers 6 CPUs/5 GPUs and diverse workloads (ResNet, BERT, Qwen2-7B), with basic ablation studies supporting design choices.

**Weaknesses:**

- Lack of novelty: Core ideas overlap with prior work (e.g., LLMPerf for LLM-based performance estimation, Moses for cross-hardware models) without clear differentiation.
- Insufficient technical details: No justification for LLM size choices, underspecified fine-tuning hyperparameters, and no clarity on how execution behavior scores map to latency.
- Experimental gaps: No tests on edge/accelerator hardware (e.g., Jetson, TPU), no comparison with modern tuners (e.g., MetaSchedule), and no sensitivity analysis for large search spaces.
- Clarity issues: Table 2 has formatting errors, Figure 4 lacks error bars, and "zero-shot" is not clearly defined.

**Questions:**

- How does your "hardware-aware reasoning" differ from LLMPerf’s prompt design?
- Can you provide full fine-tuning hyperparameters for the LLM classifier and reasoner?
- What metrics are in the parallelism/memory/computation scores, and how do they drive latency predictions?
- Will you test edge/accelerator hardware and compare with MetaSchedule?
- How do you define "zero-shot cross-hardware" (e.g., is hardware info provided manually/automatically)?

---

> ### Author Response · Authors · 2025-11-24
> **Response for Reviewer Guhc (Part #1)**
>
> We thank Reviewer for the detailed review and for recognizing the practical motivation, logical framework design, and broad experimental scope of LLMTuner. Below we address the raised concerns.
>
> > W1: “Lack of novelty” and overlap with LLMPerf / Moses
>
> > W1.1 / Q1 : Comparison with LLMPerf
>
> LLMPerf injects hardware-specific information into prompts and trains an LLM to directly regress program → latency. The core idea of LLMPerf is to learn a direct mapping from program text (augmented with device details) to latency through supervised prediction.
>
> In contrast, LLMTuner performs structured, multi-dimensional reasoning to infer interpretable execution behavior scores rather than predicting latency directly. The core advantage of this methodology is its superior generalization and transferability. While LLMPerf treats the LLM as a black-box regressor and remains limited to single-hardware tasks, our approach enables the LLM to capture unified execution behaviors that transfer across diverse architectures. This allows LLMTuner to scale effectively across 11 diverse platforms (6 CPUs + 5 GPUs) , whereas LLMPerf can only work on one platform.
>
> > W1.2: Comparison with Moses
>
> The core idea of Moses is to identify hardware-agnostic transferable features based on the lottery-ticket hypothesis and then apply domain adaptation to fine-tune these (manually designed) features to each new hardware platform. Moses still relies on feature engineering pipelines and re-adaptation when targeting new devices.
>
> In contrast, LLMTuner introduces a new paradigm: The LLM itself is prompted to reason a multi dimensional execution-behavior scores, eliminating the need for manually engineered features or domain adaptation. These scores are inherently cross-hardware because they captures execution semantics (e.g., locality, parallelism efficiency) rather than program-specific statistics.
>
> Therefore, while Moses focuses on improving feature transferability via domain adaptation, LLMTuner fundamentally replaces feature engineering with LLM-driven behavioral reasoning, achieving stronger generalization with a completely different technical route.
>
> > W2/Q2: Technical details
>
> All technical details are provided in Appendix Section C(also, anonymous source code is provided).

---

> ### Author Response · Authors · 2025-11-24
> **Response for Reviewer Guhc (Part #2)**
>
> > Q3: Score Details
>
> | Dimension       | Metric Name                      | CPU Focus                                  | GPU Focus                                               | Score Range                             |
> | --------------- | -------------------------------- | ------------------------------------------ | ------------------------------------------------------- | --------------------------------------- |
> | **Parallelism** | `core_sm_utilization_score`      | Multi-core/thread distribution             | SM coverage by grid                                     | 0–1                                     |
> |                 | `instruction_parallelism_score`  | ILP, branch prediction                     | Warp divergence                                         | 0–1                                     |
> |                 | `vectorization_occupancy_score`  | SIMD/AVX/AMX utilization                   | Occupancy (register/shared memory)                      | 0–1                                     |
> |                 | `load_balance_score`             | Thread workload balance                    | Block/warp distribution balance                         | 0–1                                     |
> |                 | `sync_barrier_score`             | Lock/barrier/atomic cost                   | `__syncthreads()`, atomic cost                          | 0–1 (higher = lower overhead)           |
> |                 | `instruction_cache_branch_score` | I-cache hit rate, branch prediction        | Kernel instruction cache usage, control flow divergence | 0–1                                     |
> | **Memory**      | `cache_global_efficiency_score`  | L1/L2/L3 hit rate, cache line utilization  | Global memory coalescing, L2 hit rate                   | 0–1                                     |
> |                 | `memory_bandwidth_util_score`    | Memory bandwidth vs peak                   | DRAM bandwidth vs peak                                  | 0–1                                     |
> |                 | `local_shared_memory_score`      | NUMA locality, LLC usage                   | Shared memory usage & conflicts                         | 0–1                                     |
> |                 | `register_pressure_level`        | Spill to stack/memory                      | Spill to local memory                                   | Level 0–4                               |
> |                 | `cache_bank_conflict_score`      | Cache line misalignment, false sharing     | Shared memory bank conflict                             | 0–1                                     |
> |                 | `data_locality_score`            | Temporal/spatial locality, prefetch effect | Global/local/shared memory access locality              | 0–1                                     |
> | **Computation** | `compute_utilization_score`      | ALU/FPU/FMA/VNNI utilization               | CUDA cores/Tensor cores/SFU utilization                 | 0–1                                     |
> |                 | `loop_unrolling_effect_score`    | Effect on ILP/pipeline                     | Effect on warp utilization                              | -1 – +1                                 |
> |                 | `bound_type_score`               | Memory-bound vs compute-bound              | Memory-bound vs compute-bound                           | 0–1 (0=memory, 0.5=balanced, 1=compute) |
> |                 | `instruction_mix_score`          | Balance of int/float/special instructions  | FMA/Tensor Core/SFU utilization mix                     | 0–1                                     |

---

> ### Author Response · Authors · 2025-11-24
> **Response for Reviewer Guhc (Part #3)**
>
> > W3/Q4:  Edge Hardware, Meta schedule, Search Space
>
> - Coverage of Edge Hardware
>
>   We respectfully clarify that our evaluation already covers representative edge architectures. Specifically, we evaluated the ARM Cortex-A76, a flagship mobile/edge SoC core , and the NVIDIA RTX 4070, typical for edge workstations.
>
>   While we could not include a specific Jetson board due to hardware access constraints, our dataset spans 11 diverse platforms, significantly broader than prior works (typically 3-5). The strong generalization on the Cortex-A76 (Table 3 ) serves as robust evidence of transferability to other ARM-based edge devices.
>
> - Comparison with MetaSchedule
>
>   Below we provide a comparison of the final best program latency found by both tuning systems on the NVIDIA A100, using precision FP32. LLMTuner consistently guides the search toward lower latency programs, proving its effectiveness as the predictive core.
>
>   | **Workload**  | **Input Shape**  | **MetaSchedule** | **LLMTuner** |
>   | ------------- | ---------------- | ---------------- | ------------ |
>   | **ResNet50**  | (1, 3, 224, 224) | 1.57ms           | 1.42ms       |
>   | **BERT-base** | (1, 128)         | 3.68ms           | 3.65ms       |
>
> - Sensitivity to Large Search Spaces
>
>   Our coarse-to-fine framework  is designed to handle massive search spaces efficiently. We include DeepSeek-V3 and Qwen2-7B (Section 5.2, Figure 4)  for our tuning experiments. These LLM workloads generate far larger and more complex search spaces, because their architectures involve longer sequence lengths and dense attention blocks that introduce a combinatorial explosion of scheduling choices compared to traditional ResNet-like models.
>
>   Table 5 also provides an explicit sensitivity analysis on the stage 1 candidate set size (K). Results show LLMTuner maintains robust accuracy (Top-5 ~0.976) even as the candidate pool expands
>
> > W4/Q5: Clarity issues
>
> Thanks for pointing this out, we will correct the fig/tab formatting in the final version.
>
> For the "zero-shot cross-hardware", it represents it refers to evaluating LLMTuner on a hardware platform not seen during training, where the model is given only the hardware name and no hardware-specific features. For example, we train on TenSet data from K80/T4 and evaluate directly on A100/H100/RTX4090 without any retraining or adaptation.

---

### Official Review · Reviewer_zf6E · 2025-11-01

**Soundness:** 2
**Presentation:** 3
**Contribution:** 2
**Rating:** 2
**Confidence:** 4

**Summary:**

This paper discusses about using LLMs to autotune tensor programs. It uses a two stage approach, where it first uses a lightweight LLM classifier to filter out poor candidates and then uses a fine-tuned LLM reasoner to analyze execution behavior and predict latency. It uses a well-established dataset in the domain (Tenset) for its experiments and claims this approach is superior to the existing solutions for training cost models.

**Strengths:**

1. The direction of work discussed here is timely and interesting.
2. The paper is written well and easier to understand the content.
3. It claims to achieve superior results outperforming SOTA LLMs and prior cost models. (Experiments demonstrate that LLMTuner significantly improves estimation accuracy by up to 64.8%, compared with general-purpose LLMs and other cost models on benchmark datasets across 6 CPU and 5 GPU platforms)
4. The solutions presented can be extended to unseen hardware (A100 and Intel 8575)
5. It was integrated into a well known framework in the domain (TVM)

**Weaknesses:**

1. Overall, it feels like an ad hoc solution where multiple LLMs are brought together to come up with some solution rather than a carefully well thought out problem solution pair.
2. Since data collection on GPUs is fast, whether we need this kind of solution itself is a question.
3. The hardware factors are still limited to the specific architectures. Like, there are no SMs in CPUs.
4. Would have been better if numbers about finetuning overhead and resource consumptions were added comparatively with existing approaches.
5. Interpretability of the approach and formalization is questionable (added more context in the questions section)

**Questions:**

1. Do we need LLMs for this kind of solution in the first place? It is true that it can take a few days to collect data for a classical ml-based cost model. However, data collection in both CPUs and especially GPUs is pretty fast.
2. Regarding the hardware factor extraction step, how are you making sure that the LLM is not hallucinating when getting data from manuals?
3. Can you formalize the reasoning process explained in the paper mathematically? For example, how representative and explainable are the intermediate scores to the runtime performance of tensor programs? How do you validate that LLMTuner is learning new reasoning patterns?
4. Why do we need a two-step process here? Wouldn't initially pre-training the first classifier make the results biased? Did you evaluate whether the reasoning model overfits to known hardware specifications mentioned during finetuning?
5. It seems the regression model at the end is trained separately. Doesn't it break the end-to-end differentiability?
6. Will this solution work beyond CPUs and GPUs? (e.g, TPUs) Can this go across heterogeneous hardware (CPU to GPU)?
7. Did you explore other NN architectures, such as GNNs, to see their performances? I suspect they can perform well too, considering we are dealing with tensor programs here.

---

> ### Author Response · Authors · 2025-11-24
> **Response for Reviewer zf6E (Part #1)**
>
> We thank the reviewer for the detailed feedback and constructive suggestions. Below we address the concerns point-by-point.
>
> > W1: It feels like an ad hoc solution.
>
> We emphasize that LLMTuner's two-stage, coarse-to-fine design is a carefully engineered framework aimed at resolving the fundamental trade-off between cross-hardware generalization and tuning efficiency, not an ad hoc collection. The coarse-to-fine architecture is essential: the lightweight classifier ensures scalability, while the reasoner generates a hardware-agnostic intermediate representation (quantitative performance scores) to enable robust zero-shot transfer.
>
> Empirical evidence strongly validates this design. By decoupling reasoning from prediction, LLMTuner achieves a 49.2% accuracy improvement on unseen hardware. Crucially, our ablation study (Table 4) confirms the necessity of this multi-stage approach, showing that the fine-selection stage boosts Top-1 accuracy by up to 32.0% (0.604 to 0.924 on CPU), proving its critical role in performance.
>
> > W2/Q1: Data collection cost and Necessity of LLMs
>
> We respectfully clarify that constructing high-quality training datasets for cost models is computationally expensive, and the necessity of LLMs lies in their superior generalization capabilities which bypass this cost entirely.
>
> - High Cost of Data Collection: While measuring a single program is fast, building a dataset sufficient for training accurate cost models is prohibitively time-consuming. For instance, TenSet [1] reports that collecting ~52M records took several weeks on cloud clusters, and Pruner[2] notes that collecting a smaller dataset of ~1.5M programs requires over 10 days. This high latency becomes a significant bottleneck given the rapid release cycle of new hardware.
>
> [1]Pruner: A Draft-then-Verify Exploration Mechanism to Accelerate Tensor Program Tuning（ASPLOS 25）
>
> [2]Tenset: A large-scale program performance dataset for learned tensor compilers(NeurIPS 21)
>
> - Necessity of LLMs (Zero-Shot Generalization): The unique value of LLMTuner is Zero-Shot Cross-Hardware Generalization, which eliminates the need for data collection on new hardware altogether. By leveraging LLM reasoning, our method achieves a 49.2% accuracy improvement on unseen hardware compared to traditional models and improves practical tuning efficiency by up to 3.27x, demonstrating a clear advantage over the "collect-then-train" paradigm.
>
> > W3:  Hardware factors limit
>
> We fully acknowledge the architectural differences between CPUs and GPUs. Our system addresses this through **Architecture-Aware Extraction** and **Unified Reasoning**:
>
> - **Extraction:** The module extracts architecture-specific factors (e.g., SM counts/Warp size for GPUs vs. SIMD/AVX widths for CPUs).
> - **Unification:** The *LLM-Reasoner* bridges the gap by mapping these disparate factors to **unified execution behavior scores** (Parallelism, Memory, Computation). For instance, it reasons how CPU SIMD width or GPU SM counts impacts vectorization efficiency, normalizing them into a comparable score space.

---

> ### Author Response · Authors · 2025-11-24
> **Response for Reviewer zf6E (Part #2)**
>
> > W4: Finetuning Overhead and Resource Consumption
>
> We acknowledge the reviewer's concern regarding the resource consumption and fine-tuning overhead of LLMTuner compared to conventional cost models. We emphasize that this higher initial cost is a strategic investment to enable Zero-Shot generalization and yield significant long-term efficiency gains.
>
> Specifically, the fine-tuning process requires a considerable resource commitment, utilizing 8x A100 GPUs for approximately 15 hours. However, this substantial upfront cost is rapidly amortized across numerous hardware platforms, as it eliminates the need for repetitive data collection and retraining cycles on every new device, achieving zero marginal cost for new hardware deployment.
>
> | **Metric**                           | **Conventional Cost Model (e.g., MLP)**            | **LLMTuner (LLM-based)**                     | **Our Advantage**                                            |
> | ------------------------------------ | -------------------------------------------------- | -------------------------------------------- | ------------------------------------------------------------ |
> | **Initial Training Resource**        | single consumer card                               | 8x A100 GPUs                                 | **Acceptable for High Value:** The resource intensive step enables a capability (Zero-Shot) that low-end models cannot achieve. |
> | **Single-Time Fine-tuning**          | ~ 2 hours                                          | ~ 15 hours                                   | **Amortized Cost:** High initial time is offset by eliminating $N$-times retraining cycle. |
> | **Total Cost Scaling (N platforms)** | Nx2 hours (Retrain required for each new hardware) | ~15 hours (Train once, Zero-Shot deployment) | **Long-Term Savings:** zero cost for new hardware deployment. |
> | **Tuning Efficiency**                | 1.0x (Baseline)                                    | ~ 3.27x                                      | **Performance Gain:** Faster convergence reduces costly real-hardware measurement time (Figure 4). |
> | **Inference Cost Mitigation**        | N/A                                                | Coarse-to-Fine Structure                     | **Resource Management:** Only applies high-cost LLM reasoning to a small subset (e.g., Top-32) of candidates. |
>
> > W5 & Q3: Formalization, Interpretability, and Validation
>
> - Mathematical Formalization: We formalize the performance modeling as a two-stage composite function:(1)Structured Reasoning: An LLM $\mathcal{M}$ acts as a semantic feature extractor mapping the program $p$ and hardware specifications $h$ to a quantitative score vector $S$ and reasoning trace $R$.  $S, R = \mathcal{M}(p, h)$, where $S \in [0,1]^d$ represents normalized metrics across dimensions.(2)Latency Regression: A lightweight MLP $f_{reg}$ maps the semantic scores to predicted latency $\hat{T}$, $\hat{T} = f_{reg}(S)$.
>
> - Interpretability and Validation
>
>   Unlike prior black-box embedding methods, LLMTuner produces human-readable diagnostics. For a Tiled MatMul on A100, it correctly separates conflicting performance factors:
>
>   **`cache_global_efficiency_score`**: 0.9 — Global loads are vectorized (`float4`) and fully coalesced.
>
>   **`cache_bank_conflict_score`**: 0.2 — A stride of 64 causes severe shared-memory bank conflicts due to inadequate padding.
>
>   This example shows LLMTuner identifies that the kernel is memory-bound by bank conflicts despite perfect global coalescing, demonstrating learned causal hardware reasoning. This capability underlies our 49.2% accuracy improvement on unseen hardware.

---

> ### Author Response · Authors · 2025-11-24
> **Response for Reviewer zf6E (Part #3)**
>
> -
>
> > Q2: Preventing Hallucination in Hardware Extraction
>
> We avoid hallucination by using a fixed JSON template with predefined hardware fields, which restricts the LLM to filling in values rather than inventing new attributes. This extraction step is independently testable: we validate all extracted values against ground-truth manuals, and in practice the extracted hardware parameters consistently match the official specifications.
>
> > Q4: Two-Stage Design / Classifier bias / Overfitting Concerns
>
> - Two stages: The coarse-to-fine design improves scalability: the lightweight Classifier quickly removes clearly poor candidates, allowing the heavier Reasoner to focus only on a small promising set.
> - Classifier bias: Experimental results show no measurable bias. As shown in Tab 4, in Stage-1 alone, the classifier achieves consistent Top-1 accuracy across GPU (0.765) and CPU (0.604) platforms, and after applying Stage-2, the relative ranking of surviving candidates remains stable across hardware (e.g., GPU: 0.765→0.919, CPU: 0.604→0.924).
> - Overfitting: The Reasoner outputs hardware-agnostic behavior scores, and its strong accuracy on *unseen hardware* (A100, H100, Intel 8575C) demonstrates that it does not overfit to hardware specifications used in fine-tuning.
>
> > Q5: It seems the regression model at the end is trained separately. Doesn't it break the end-to-end differentiability?
>
> LLMTuner is not intended to be an end-to-end differentiable model. As described in Section 4.3 and Appendix C.4, the LLM Reasoner outputs hardware-agnostic execution-behavior scores, and the regression model is only a lightweight post-processing module that maps these scores to latency. Training it separately does not affect the pipeline’s correctness or performance, and LLMTuner still achieves strong accuracy (e.g., Top-1 0.919 on K80).
>
> > Q6: Will this solution work beyond CPUs and GPUs? (e.g, TPUs) Can this go across heterogeneous hardware (CPU to GPU)?
>
> Our method is principally hardware-agnostic: LLMTuner only relies on structured hardware descriptors and performs execution-behavior reasoning at an abstract level. In this sense, the approach can in principle be applied to TPUs or other accelerators, provided that their hardware factors are available. Due to the lack of TPU access, however, we are currently unable to include TPU experiments.
>
> The goal of LLMTuner is intra-family generalization rather than cross-family transfer. Regarding heterogeneous transfer (e.g., CPU→GPU), CPU and GPU architectures differ dramatically in both performance characteristics and program representations, so a model trained solely on CPU traces is not expected to perform well on GPUs without additional adaptation. Therefore, we intentionally restrict our evaluation to within-family scenarios, where the underlying execution patterns are more consistent. Within this intended scope, LLMTuner demonstrates strong zero-shot generalization—for example, across GPUs (Nvidia K80 → A100/H100) and across CPUs (Intel 8276 → AMD Cortex)—indicating that the approach is robust within each hardware family.
>
> > Q7: Exploration other NN architectures, such as GNNs.
>
> As discussed in the Related Work section, prior models based on GNNs or other NN architectures (MLPs, Transformers) follow the same fundamental paradigm: they directly learn a program → latency mapping. This paradigm inherently lacks cross-hardware generalization, because the learned representation captures only program-level features while ignoring how the program actually executes on different hardware.
>
> In contrast, LLMTuner models the execution behavior of the program on a given device using unified behavior score. This hardware–program interaction is essential for generalization across devices.

---

### Official Review · Reviewer_EMgf · 2025-11-02

**Soundness:** 2
**Presentation:** 3
**Contribution:** 2
**Rating:** 4
**Confidence:** 3

**Summary:**

This paper proposes LLMTuner, a framework using LLMs to predict tensor program performance by analyzing unified execution behaviors (parallelism, memory, computation) instead of hand-crafted features. It employs a two-stage approach: coarse filtering via LLM classifier and fine ranking via a distilled LLM reasoner that generates scores for regression-based latency prediction. Experiments show 64.8% accuracy improvement on seen hardware and 49.2% on unseen hardware.

**Strengths:**

1. The shift from hand-crafted, hardware-specific features to LLM-based unified execution behavior analysis tackles a significant pain point in tensor program tuning—poor cross-platform generalization of traditional cost models. This is a meaningful direction that addresses real engineering challenges.

2. The two-stage coarse-to-fine design achieves a good balance between accuracy and efficiency. Knowledge distillation from DeepSeek-R1 to Qwen2.5-7B reduces inference time by 30 times while maintaining comparable performance, demonstrating practical applicability for real-world deployment.

**Weaknesses:**

1. The core technical contribution is essentially fine-tuning LLMs with structured prompts and supervised distillation from DeepSeek-R1, followed by a regression model on top of the LLM outputs. While the application domain is novel, the methodology itself (prompt engineering + distillation + regression) is relatively incremental and lacks fundamental innovation in either the LLM techniques or the performance modeling approach.

2. The generalization claims are a bit overstated in two ways. First, the "unseen hardware" in Table 3 (A100, H100, RTX 4070) all share the CUDA architecture with the training platforms (K80, T4). True zero-shot generalization should be demonstrated on fundamentally different ISAs like AMD GPUs, Apple Silicon, or Google TPU. Second, the LLMTuner-Reasoner outputs scores for what appears to be a fixed set of metrics like SM utilization and warp efficiency. It's unclear whether these metrics are hardware-adaptive or remain static across all platforms. For example, Blackwell architecture supports native FP4 Tensor Cores while earlier architectures don't—does the LLM understand such hardware-specific capabilities? How does a fixed metric schema handle the fundamental differences between NVIDIA's streaming multiprocessors, AMD's compute units, and Apple's neural engine? The paper doesn't clarify whether the evaluation schema varies by hardware or if it assumes a one-size-fits-all metric set, which undermines the claim of hardware-agnostic reasoning.

3. Section C.3 reveals a concerning data leakage issue. The paper states that "real latency measurements are provided as additional input to DeepSeek-R1 to guide its reasoning" during training data generation. This creates a problematic chain: DeepSeek-R1 receives ground-truth latency hints, generates reasoning and scores based on these hints, then Qwen2.5-7B learns from this distillation data, and finally the regression model is trained on these scores paired with the same ground-truth latencies. Even though Listing 1 shows the hint is not explicitly included in the output, the reasoning and scores may still implicitly encode latency information rather than representing genuine program-hardware analysis. The paper provides no ablation to quantify what percentage of the 100K training samples used hints, nor does it compare performance when trained purely on hint-free reasoning data. This makes it impossible to assess whether the model's strong performance comes from learned reasoning patterns or from memorizing latency-conditioned outputs.

**Questions:**

1. The paper claims to use a structured multi-dimensional evaluation schema but Appendix E only shows a partial example with 3 sub-metrics. Can you provide the complete metric schema with all sub-metrics used in practice? More importantly, how does this schema adapt to different hardware architectures? Even within the same vendor, GPU architectures have significant differences—for instance, Blackwell supports native FP4 Tensor Cores, Hopper supports FP8, while Ampere supports neither. Does the LLM understand such generation-specific capabilities, or does it treat all NVIDIA GPUs uniformly? Beyond NVIDIA, how do you handle metrics for platforms with fundamentally different designs—NVIDIA SMs versus AMD CUs versus Apple Neural Engine? Are the metrics hardware-adaptive or do you use a fixed schema across all platforms?

2. Regarding the latency hints mentioned in Section C.3, what percentage of the 100K distillation samples actually used ground-truth latency as hints? This is critical for understanding potential data leakage. Can you provide an ablation comparing LLMTuner-Reasoner's Top-1 and Top-5 accuracy when trained on distillation data generated with hints versus without hints?

---

> ### Author Response · Authors · 2025-11-24
> **Response for Reviewer EMgf (Part #1)**
>
> We thank the reviewer for the detailed feedback and constructive suggestions. Below we address the concerns point-by-point.
>
> > W1: Method is incremental.
>
> We appreciate the reviewer’s perspective, and we would like to clarify that the core contribution of our work is not introducing a new LLM training trick, but rather establishing a new modeling paradigm for performance estimation in tensor program tuning.
>
> Prior approaches, whether using MLPs, or Transformers, ultimately rely on the same “program → latency” regression formulation. Such models only learn program-level patterns and thus struggle to generalize across hardware, as they have no mechanism to capture how a given device actually executes a program. LLMTuner replaces this paradigm with a behavior-centric formulation: “program + hardware → execution behavior → latency”.
>
> Central to this shift is using the LLM not as a latency regressor, but as a structured execution-behavior analyzer. Base on this insight，our two-stage coarse-to-fine design naturally emerges: a lightweight classifier filters poor candidates, and the Reasoner performs structured behavior analysis.
>
> In summary, the novelty lies in introducing this hardware-grounded execution-behavior modeling paradigm, which is fundamentally different from prior feature-based or direct-regression approaches.

---

> ### Author Response · Authors · 2025-11-24
> **Response for Reviewer EMgf (Part #2)**
>
> > W2/Q1: Hardware Generalization & Metric Schema
>
> We use a fixed 16-dimensional execution-behavior schema spanning parallelism, memory efficiency, and computation. A key clarification is that while the dimensionality is fixed, the semantics of each dimension are hardware-adaptive: the schema does not encode NVIDIA-specific assumptions; instead, each score is interpreted relative to the hardware specifications supplied to the model.
>
> For example, `vectorization_occupancy_score` reflects SIMD/AVX/AMX utilization on CPUs but captures warp occupancy, register pressure, and shared-memory constraints on GPUs; similarly, computation-related metrics naturally adjust for future GPU architectures (e.g., FP4/FP8 tensor-core throughput) when such capabilities are present in the provided specs. This design makes LLMTuner hardware-agnostic in framework yet hardware-aware through input, enabling consistent reasoning across diverse CPU/GPU architectures without redesigning the schema.
>
> While our current experiments focus on CPU and NVIDIA GPU platforms due to available measurement data, the framework itself readily extends to AMD, Apple, or TPU architectures because the schema operates at a hardware-neutral abstraction level and is instantiated directly from device descriptions.
>
> The full score dimension is provided below：
>
> | Dimension       | Metric Name                      | CPU Focus                                  | GPU Focus                                               | Score Range                             |
> | --------------- | -------------------------------- | ------------------------------------------ | ------------------------------------------------------- | --------------------------------------- |
> | **Parallelism** | `core_sm_utilization_score`      | Multi-core/thread distribution             | SM coverage by grid                                     | 0–1                                     |
> |                 | `instruction_parallelism_score`  | ILP, branch prediction                     | Warp divergence                                         | 0–1                                     |
> |                 | `vectorization_occupancy_score`  | SIMD/AVX/AMX utilization                   | Occupancy (register/shared memory)                      | 0–1                                     |
> |                 | `load_balance_score`             | Thread workload balance                    | Block/warp distribution balance                         | 0–1                                     |
> |                 | `sync_barrier_score`             | Lock/barrier/atomic cost                   | `__syncthreads()`, atomic cost                          | 0–1 (higher = lower overhead)           |
> |                 | `instruction_cache_branch_score` | I-cache hit rate, branch prediction        | Kernel instruction cache usage, control flow divergence | 0–1                                     |
> | **Memory**      | `cache_global_efficiency_score`  | L1/L2/L3 hit rate, cache line utilization  | Global memory coalescing, L2 hit rate                   | 0–1                                     |
> |                 | `memory_bandwidth_util_score`    | Memory bandwidth vs peak                   | DRAM bandwidth vs peak                                  | 0–1                                     |
> |                 | `local_shared_memory_score`      | NUMA locality, LLC usage                   | Shared memory usage & conflicts                         | 0–1                                     |
> |                 | `register_pressure_level`        | Spill to stack/memory                      | Spill to local memory                                   | Level 0–4                               |
> |                 | `cache_bank_conflict_score`      | Cache line misalignment, false sharing     | Shared memory bank conflict                             | 0–1                                     |
> |                 | `data_locality_score`            | Temporal/spatial locality, prefetch effect | Global/local/shared memory access locality              | 0–1                                     |
> | **Computation** | `compute_utilization_score`      | ALU/FPU/FMA/VNNI utilization               | CUDA cores/Tensor cores/SFU utilization                 | 0–1                                     |
> |                 | `loop_unrolling_effect_score`    | Effect on ILP/pipeline                     | Effect on warp utilization                              | -1 – +1                                 |
> |                 | `bound_type_score`               | Memory-bound vs compute-bound              | Memory-bound vs compute-bound                           | 0–1 (0=memory, 0.5=balanced, 1=compute) |
> |                 | `instruction_mix_score`          | Balance of int/float/special instructions  | FMA/Tensor Core/SFU utilization mix                     | 0–1                                     |
>
> >

---

> ### Author Response · Authors · 2025-11-24
> **Response for Reviewer EMgf (Part #3)**
>
> > W3/Q2: Latency Hints and Potential Data Leakage
>
> - Clarification: Latency Hints Improve Teacher Reasoning Quality, Not Student Input
>
> We thank the reviewer for the thoughtful question. However, we clarify that the Student Model(LLMTuner-Reasoner) never sees the latency hints, and therefore no data leakage occurs. The latency hints are only visible to the Teacher (DeepSeek-R1) during the data generation phase. The Student model (LLMTuner-Reasoner) receives only the program code and hardware specs as input.
>
> - The Role of Latency Hints: Preventing Teacher Hallucination
>
> We introduced hints to solve a specific hallucination problem in the Teacher Model. Without ground-truth guidance, even powerful LLMs like DeepSeek-R1 sometimes generate "plausible but incorrect" reasoning. For example, for a program that is empirically slow, the Teacher might hallucinate that it is "highly efficient" because it misinterprets a specific loop structure. By providing the latency as a "hint" (e.g., "this program is slow"), we force the Teacher to look deeper and find the true bottleneck (e.g., "low cache hit rate") that explains the poor performance. This ensures the training dataset contains high-quality, physically grounded reasoning traces. 3. Dataset Composition
>
> Approximately 30% of the 100K distillation samples (29,842 out of 99,732 samples) use hints. We observed the Teacher performs well on easy cases (highly efficient or highly inefficient programs) without hints but struggles on harder ones. Therefore, hints are applied adaptively only to difficult samples to ensure reasoning quality while preserving dataset diversity.
>
> - Ablation Study
>
> To empirically prove that our performance comes from learned reasoning and not leakage, we conducted the ablation study suggested by the reviewer. We trained two versions(w/ and w/o Hints) of LLMTuner-Reasoner and test it on the TenSet K80 dataset :
>
> | **Model**         | **Top-1 Accuracy** | **Top-5 Accuracy** |
> | ----------------- | ------------------ | ------------------ |
> | trained w/o Hints | 0.833              | 0.902              |
> | trained w/ Hints  | 0.919              | 0.976              |
>
> The results show that incorporating latency hints during the teacher's data generation phase significantly improves the student's final performance.

---

### Author Response · Authors · 2025-11-28
**Global Response**

We thank all reviewers for their time and constructive feedback. We are encouraged that they recognized our novel cross-hardware execution-behavior modeling paradigm (Reviewers EMgf, wniB), the effectiveness of our coarse-to-fine framework (Reviewers EMgf, Guhc, DUzA), and the strong empirical performance across multiple hardware targets (Reviewers zf6E, wniB, Guhc, DUzA).

Across all reviews, we clarified LLMTuner’s core contribution: a behavior-centric cross-hardware execution-reasoning paradigm that generalizes across unseen devices through a unified 16-dimensional hardware-adaptive metric schema. We provided new ablations, experiments, and comparisons addressing specific concerns:

- **Reviewer EMgf**:
    - Clarified the paradigm innovation (program + hardware → execution behavior → latency).
    - Provided the full 16-D metric schema with detailed interpretation.
    - Added a latency-hint ablation (Top-1: 0.833→0.919), demonstrating the gains come from better teacher reasoning rather than leakage.
- **Reviewer zf6E**:
    - Emphasized LLMTuner’s zero-shot cross-hardware generalization, eliminating hardware-specific data collection.
    - Justified the two-stage design with up to 32% Top-1 improvement and interpretable score vectors.
    - Verified robustness against overfitting and reported competitive overhead relative to conventional models.
- **Reviewer Guhc**:
    - Clarified the differences from prior works LLMPerf (black-box regression) and Moses (manual features).
    - Provided the full 16-D schema and explained its technical role.
    - Added results on edge hardware and a direct comparison showing that LLMTuner yields lower-latency schedules than MetaSchedule on A100.
- **Reviewer wniB**:
    - Clarified the removal of hand-crafted hardware features and the automatic inference of high-level dimensions.
    - Justified the absence of a separate communication dimension.
    - Added multi-dimensional ablations showing that all three dimensions jointly drive accuracy.
- **Reviewer DUzA**:
    - Added new results within MetaSchedule (FP16, Tensor Cores), unseen ops/shapes generalization, and faster search on A100.
    - Reaffirmed auto-tuning’s relevance for fusion, new hardware, and subgraph optimization.
    - Added comparisons with cuDNN/cuBLAS and Triton, showing that LLMTuner delivers superior performance in end-to-end, fused, and non-template workloads.

We hope that the additional experiments and clarifications further strengthen the paper and address the reviewers’ concerns.

---

### Author Response · Authors · 2025-12-02
**Summary of Review Discussion**

Dear ACs and Reviewers,

Thank you for your valuable contributions to our work. To assist the newly assigned AC and help reduce their workload, we provide a concise summary of the key points from the review process.

---
**Strength.** Our work presents LLMTuner, a novel LLM-based framework that enables zero-shot cross-hardware performance modeling for tensor program tuning. We appreciate the reviewers’ thoughtful assessments and their recognition of the notable strengths of our work. Specifically:
- **Novel cross-hardware modeling paradigm**: Reviewers acknowledged our shift from hand-crafted hardware-specific features to a unified, LLM-driven execution-behavior analysis (EMgf, zf6E, Guhc, wniB).
- **Strong empirical performance**: LLMTuner achieves up to 64.8% accuracy improvement on seen hardware and 49.2% on unseen hardware, while improving tuning efficiency by up to 3.27× (zf6E, Guhc, wniB).
- **Coarse-to-fine design**: The two-stage framework balances efficiency and accuracy, and is validated through ablations showing up to 32% Top-1 improvement (EMgf, zf6E, wniB).
---
**Concerns and Our Responses.** During the discussion period, we clarified reviewers’ misunderstandings and addressed concerns about our method and experiments. Due to the OpenReview issue, further discussion was not possible, so we summarize our responses here:
- **Misunderstandings about technological novelty**. The use of LLMs with prompting + distillation + regression was seen as incremental. (EMgf, zf6E, Guhc)
    - **Our Addressing**: We clarified that the **core novelty** is not in LLM training tricks, but in introducing a **behavior-centric modeling paradigm**: “program + hardware → execution behavior → latency”, replacing the traditional “program → latency” regression. This enables zero-shot generalization across hardware without retraining or feature engineering.
- **Concerns about generalization to heterogeneous hardware.** Reviewers raised questions about (1) generalization across ISAs (e.g., CPU→GPU, TPU), and (2) whether the 16-D metric schema is hardware-adaptive. (EMgf, zf6E, Guhc, wniB)
    - **Our Addressing:** (1)Because CPU/GPU/TPU architectures differ greatly in performance behavior and program representations, cross-ISA transfer is not expected without additional adaptation. LLMTuner focuses on **zero-shot intra-family generalization**—e.g., a model trained on older GPUs (K80) can be directly applied to newer ones (A100, H100). Experiments show robust zero-shot generalization on 5 GPUs and 6 CPUs.(2) We provided the full **16-D metric details**, clarified its hardware-adaptive semantics, and emphasized that it is architecture-neutral while remaining hardware-aware through input specifications.
- **Concerns about more comparisons and modern context.** Reviewers argued that (1)need comparisons against cuDNN and Triton; (2)TenSet benchmark is outdated, as it is based on Ansor, and lacks Tensor Core support.(DUzA)
    - **Our Addressing**: (1) LLMTuner targets **automatic cross-platform kernel generation** tasks of Tensor Complier, which is fundamentally different in scope from **vendor libraries (cuDNN/cuBLAS) and expert-written templates(Triton)**. We provide additional experiments showing that, for implementing ResNet-50 and BERT-base, **LLMTuner outperforms cuDNN/cuBLAS** and is competitive with Triton. (2)We further integrate LLMTuner into MetaSchedule, demonstrating that our method supports FP16/Tensor Core, showing effectiveness beyond TenSet/Ansor: (i)**Generalization** to unseen operators and shapes (100% Top-1 on conv, 99.4% on matmul), (ii)**Search efficiency:** LLMTuner finds better kernels faster than MetaSchedule’s default cost model on real LLM workloads.
- **Concerns about score details, interpretability, and potential data leakage.** Reviewers asked for (1)contribution of three behavior dimensions, (2)clarification on how the scores relate to latency, and (3)potential leakage of latency hints used in the teacher reasoning into the student training. (wniB, Guhc, EMgf)
    - **Our Addressing:** (1)We provided **complete ablations**, showing that all three behavior dimensions (parallelism, memory, computation) are necessary—removing any reduces Top-1 by 8–16%. (2)We also offered **interpretable examples** to demonstrate that the inferred scores reflect how programs behave on hardware. (3)In addition, we clarified that latency hints are used **only during teacher data generation** and are **never exposed to the student**, and we included an **ablation** showing that the performance gain is driven by improved reasoning rather than leakage (Top-1: 0.833 → 0.919).
---
We have put significant effort into our responses, including new experiments, analyses, and detailed clarifications. We believe that all concerns have been thoroughly addressed. We are sincerely grateful to the reviewers for their thoughtful feedback. We extend our sincere respect and appreciation to everyone involved.

Sincerely,

Authors

---

### Meta-Review · Area_Chair_PELf · 2026-01-10

**Summary:**

We have 5 reviewers for this submission, and every reviewer contributed good comments for improving the submission. The authors have also demonstrated genuine efforts to provide answers with new experiment results to all comments.

Unfortunately, none of the reviewers have upgraded their ratings based on the rebuttal. Though I would expect some of the authors answers may have changed some reviewers’ mind and get a higher rating, the existence of two low rating (2) from two reviewers with high confidence level (4), two marginally low rating (4), and only one marginally high rating (6), I’d still suggest a reject for the current submission as is.

**Reviewer Concerns:**

The 5 reviewers raised a set of good comments for the submission, and the questions include a wide range of concerns, such as

•	The lack of novelty for the technical contribution
•	Not clear about the generalizability for different hardware architectures/features
•	Data leakage issues during training
•	Motivation is not clear
•	Lack of overhead analysis and data
•	Interpretability concerns about the solution
•	Lack of novelty compared to prior work (e.g., LLMPerf, Moses)
•	Lack of sufficient technical details
•	No test on edge devices and sensitivity analysis
•	The ignore of communication as a dimension for exploration
•	No ablation studies
•	Weak motivation on the need for LLMTuner
•	Weak evaluation on outdated dataset (TenSet) and the lack of evaluation on real workloads

I believe the authors tried to address all of those comments with some reasonable answers. But given that none reviewers raised their ratings explicitly, the answers may not be completely sufficient.

**Reviewer Scores:**

It's very hard to tell as none has expressed such inclination.

---

### Decision · Program_Chairs · 2026-01-26

Reject